# Optimal Strong Regret and Violation in Constrained MDPs via Policy Optimization

**Francesco Emanuele Stradi**
Politecnico di Milano
`francescoemanuele.stradi@polimi.it`

**Matteo Castiglioni**
Politecnico di Milano
`matteo.castiglioni@polimi.it`

**Alberto Marchesi**
Politecnico di Milano
`alberto.marchesi@polimi.it`

**Nicola Gatti**
Politecnico di Milano
`nicola.gatti@polimi.it`

## Abstract

We study online learning in *constrained MDPs* (CMDPs), focusing on the goal of attaining sublinear *strong* regret and *strong* cumulative constraint violation. Differently from their standard (weak) counterparts, these metrics do *not* allow negative terms to compensate positive ones, raising considerable additional challenges. Efroni et al. (2020) were the first to propose an algorithm with sublinear strong regret and strong violation, by exploiting linear programming. Thus, their algorithm is highly inefficient, leaving as an open problem achieving sublinear bounds by means of policy optimization methods, which are much more efficient in practice. Very recently, Müller et al. (2024) have partially addressed this problem by proposing a policy optimization method that allows to attain $\widetilde{\mathcal{O}}(T^{0.93})$ strong regret/violation. This still leaves open the question of whether *optimal* bounds are achievable by using an approach of this kind. We answer such a question affirmatively, by providing an efficient policy optimization algorithm with $\widetilde{\mathcal{O}}(\sqrt{T})$ *strong* regret/violation. Our algorithm implements a primal-dual scheme that employs a state-of-the-art policy optimization approach for adversarial (unconstrained) MDPs as primal algorithm, and a UCB-like update for dual variables.

## 1 Introduction

*Markov decision processes* (MDPs) (Puterman, 2014) have emerged as the most natural way of modeling learning problems in which an agent sequentially interacts with an (unknown) environment. In MDPs, the goal is to learn an action-selection policy that maximizes learner's cumulative rewards. However, in many real-world applications of interest, there are some additional requirements that the learner has to meet while learning. For instance, in autonomous driving one has to avoid colliding with obstacles (Wen et al., 2020; Isele et al., 2018), in ad auctions the allocated budget must *not* be depleted (Wu et al., 2018; He et al., 2021), while in recommendation system offending items should *not* be presented to users (Singh et al., 2020). In order to capture requirements of this kind, *constrained MDPs* (CMDPs) have been introduced (Altman, 1999). These models augment classical MDPs by adding agent's costs that the learner is constrained to keep below some given thresholds.

In this paper, we study *online learning* problems in *finite-horizon episodic* CMDPs. In such settings, the learner's goal is twofold. On the one hand, they aim at minimizing their cumulative *regret*, which measures how much they lose over the episodes compared to always playing an optimal (*i.e.*, reward-maximizing) policy that satisfies the constraint. On the other hand, the learner also wants to ensure that cost constrains are satisfied during learning. This is measured by means of the cumulative *constraint violation* (or simply violation for short), which measures by how much agent's costs exceed their respective thresholds, cumulatively over the episodes. Ideally, the learner would like that both regret and violation grow sublinearly in the number of episodes $T$ of the CMDP.

The classical notions of regret and violation are usually called *weak*, due to the fact that they allow for negative terms to cancel out positive ones. In CMDPs, this means that the (weak) regret can

be easily controlled by using policies achieving large rewards *without* satisfying the constraints. Similarly, the (weak) violation can be controlled by adopting policies satisfying cost constraints by a large margin. However, this behavior is most of the times unacceptable in real-world applications. For instance, in autonomous driving, the learner does *not* have the option of being overly safe in some episodes so as to compensate for crashes occurred in previous episodes.

The *strong* regret and the *strong* constraint violation are much more reasonable metrics compared to their weak counterparts, as they do *not* allow negative terms to cancel out positive ones. However, achieving sublinear strong regret/violation in CMDPs is much more challenging.

Efroni et al. (2020) were the first to provide a learning algorithm with (optimal) $\widetilde{\mathcal{O}}(\sqrt{T})$ strong regret/violation in general CMDPs. However, their algorithm works by solving linear programs defined over the space of occupancy measures, a task that is highly inefficient in practice. Ideally, one would like learning algorithms that avoid dealing with occupancy measures, by directly optimizing over the policy space. Such policy optimization algorithms are much more efficient and desirable in practice. By leveraging a primal-dual scheme, Efroni et al. (2020) designed a first policy optimization algorithm for CMDPs, though it can only achieve sublinear *weak* regret and *weak* violation, leaving as an open problem whether an analogous result is achievable for the strong metrics.

Very recently, (Müller et al., 2024) partially addressed this problem by proposing a primal-dual policy optimization algorithm attaining $\widetilde{\mathcal{O}}(T^{0.93})$ *strong* regret and *strong* violation. However, the bounds achieved by such an algorithm remain largely suboptimal, leaving a big gap that still needs to be closed.

In this paper, we answer the following question left open by (Efroni et al., 2020; Müller et al., 2024):

> *Is it possible to achieve **optimal** $\widetilde{\mathcal{O}}(\sqrt{T})$ bounds on the **strong** regret and the **strong** constraint violation in CMDPs by using an **efficient** primal-dual **policy optimization** algorithm?*

We answer the question above affirmatively. To do so, we design a learning algorithm that exploits a novel primal-dual scheme. Specifically, our algorithm adopts, as primal regret minimizer, a state-of-the-art policy optimization algorithm for adversarial (unconstrained) MDPs, while it leverages an approach based on upper confidence bounds in order to build a dual regret minimizer. Crucially, the updates of dual variables performed by our algorithm do *not* resort to optimizing over the space of occupancy measures, making our algorithm a fully policy optimization approach, and, thus, efficient.

## 2 RELATED WORKS

Online learning (Cesa-Bianchi & Lugosi, 2006; Orabona, 2019) in MDPs has received considerable attention over the last decade. Specifically, online MDPs have been studied both in stochastic settings, namely, when the rewards are sampled from a fixed distribution (see, *e.g.*, (Auer et al., 2008; Even-Dar et al., 2009)) and in adversarial ones, *i.e.*, when no statistical assumption is made on the rewards (see, *e.g.*, (Neu et al., 2010; Rosenberg & Mansour, 2019b;a; Jin et al., 2020)).

Online learning in CMDPs has generally been studied in settings with stochastic rewards and constraints. Zheng & Ratliff (2020) deal with fully-stochastic episodic CMDPs, assuming known transitions. Efroni et al. (2020) propose two approaches (and four algorithms) to deal with the exploration-exploitation trade-off in episodic CMDPs. The first approach (*i.e.*, the first two algorithms) resorts to a linear programming formulation of CMDPs and obtains sublinear *strong* regret and *strong* constraint violation. The second approach (*i.e.*, the last two algorithms) relies on a primal-dual (or dual) formulation of the problem, guaranteeing sublinear (weak) regret and (weak) constraint violation, *i.e.*, allowing for cancellations. Finally, very recently Müller et al. (2024) proposed the first primal-dual method capable of achieving sublinear *strong* regret and *strong* constraint violation. The algorithm employs a policy optimization update and a regularized Lagrangian formulation of CMDPs in order to attain $\widetilde{\mathcal{O}}(T^{0.93})$ *strong* regret and *strong* violation bounds.

Finally, it is worth remarking that online CDMPs have also been addressed in adversarial settings. Precisely, (Wei et al., 2018; Qiu et al., 2020; Stradi et al., 2024b) address CMDPs with adversarial losses, but they only provide guarantees in terms of (weak) constraint violations, thus allowing for cancellations. Very recently, Stradi et al. (2024a) study CMDPs with adversarial losses and provide

guarantees on *strong* constraint violation. Nevertheless, their approach is *not* primal-dual and their algorithm has to (inefficiently) solve a convex program at each episode.

Due to space constraints, we refer the reader to Appendix A for further details on related works.

## 3 PRELIMINARIES

In this section, we introduce notation and all the definitions needed in the rest of the paper.

### 3.1 CONSTRAINED MARKOV DECISION PROCESSES

A CMDP (Altman, 1999) is defined as a tuple $(X, A, m, P, r, G, \alpha)$, where:

- $X$ and $A$ are finite state and action spaces, respectively.
- $m \in \mathbb{N}_{>0}$ is the number of constraints.
- $P : X \times A \times X \to [0, 1]$ is a transition function, with $P(x'|x, a)$ denoting the probability of going from state $x \in X$ to $x' \in X$ by taking action $a \in A$.[1]
- $r \in [0, 1]^{|X \times A|}$ is a reward vector, whose component $r(x, a)$ encodes the reward obtained by the learner when taking action $a \in A$ in state $x \in X$.
- $G \in [0, 1]^{|X \times A| \times m}$ is a cost matrix stacking a cost vector $g_i \in [0, 1]^{|X \times A|}$ for each constraint $i \in [m]$.[2] The component $g_i(x, a)$ of $g_i$ encodes the cost associated with the $i$-th constraint incurred by the learner when taking action $a \in A$ in state $x \in X$.
- $\alpha \in [0, L]^m$ is a threshold vector, whose component $\alpha_i$ is for constraint $i \in [m]$.

We study CMDPs where rewards and costs are *stochastic*, namely, they are randomly sampled according to some probability distributions $\mathcal{R}$ and $\mathcal{G}_i$ whose expected values are $r$ and $g_i$, respectively. Specifically, a reward vector $\widetilde{r} \in [0, 1]^{|X \times A|}$ is sampled according to a probability distribution $\mathcal{R}$ supported on $[0, 1]^{|X \times A|}$ with $\mathbb{E}_{\mathcal{R}}[\widetilde{r}] = r$. Similarly, for every constraint $i \in [m]$, a random cost vector $\widetilde{g}_i \in [0, 1]^{|X \times A|}$ is sampled from a distribution $\mathcal{G}_i$ supported on $[0, 1]^{|X \times A|}$ with $\mathbb{E}_{\mathcal{G}_i}[\widetilde{g}_i] = g_i$.

The learner interacts with the CMDP by employing a *policy* $\pi : X \times A \to [0, 1]$, which defines a probability distribution over actions at each state. For ease of notation, we denote by $\pi(\cdot|x)$ the probability distribution for a state $x \in X$, with $\pi(a|x)$ denoting the probability of action $a \in A$. In the following, we denote by $\Pi$ the set of all the possible policies the learner can choose from.

Algorithm 1 depicts the CMDP interaction process, when the learner uses $\pi : X \times A \to [0, 1]$ and sampled reward and cost vectors are $\widetilde{r} \in [0, 1]^{|X \times A|}$ and $\widetilde{g}_i \in [0, 1]^{|X \times A|}$ for $i \in [m]$, respectively.

---

**Algorithm 1** Learner-Environment Interaction

---

**Require:** Policy $\pi : X \times A \to [0, 1]$, $\widetilde{r} \in [0, 1]^{|X \times A|}$, $\widetilde{g}_i \in [0, 1]^{|X \times A|}$ for $i \in [m]$
1: Environment initializes state to $x_0 \in X_0$
2: **for** $k = 0, \ldots, L - 1$ **do**
3:      Learner plays $a_k \sim \pi(\cdot|x_k)$
4:      Learner gets reward $\widetilde{r}(x_k, a_k)$
5:      Learner incurs cost $\widetilde{g}_i(x_k, a_k)$ for $i \in [m]$
6:      Environment evolves to $x_{k+1} \sim P(\cdot|x_k, a_k)$

---

Given a transition function $P : X \times A \times X \to [0, 1]$, a policy $\pi : X \times A \to [0, 1]$, and a generic vector $v \in [0, 1]^{|X \times A|}$ indexed on state-action pairs, we introduce a value function $V^{\pi, P}(\cdot|v) : X \to [0, L]$

---

[1] W.l.o.g., in this paper we assume to work with *loop-free* CMDPs. Formally, this means that $X$ is partitioned into $L$ layers $X_0, \ldots, X_L$ such that the first and the last layers are singletons, *i.e.*, $X_0 = \{x_0\}$ and $X_L = \{x_L\}$, and that $P(x'|x, a) > 0$ only if $x' \in X_{k+1}$ and $x \in X_k$ for some $k \in \{0, \ldots, L - 1\}$. Notice that any finite-horizon CMDP with horizon $H$ that is *not* loop-free can be cast into a loop-free one by suitably duplicating the state space $H$ times, *i.e.*, a state $x$ is mapped to a set of new states $(x, k)$, where $k \in \{1, \ldots, H\}$.

[2] In this paper, we denote with $[n] := \{1, \ldots, n\}$ the set of all the integers from 1 to $n \in \mathbb{N}_{>0}$.

that is defined as follows, for every $k \in \{0, \ldots, L-1\}$ and $x \in X_k$:

$$V^{\pi,P}(x,v) := \mathbb{E}_{\pi,P}\left[\sum_{k'=k}^{L-1} v(x_{k'}, a_{k'}) \mid x_k = x\right].$$

Moreover, we define $V^{\pi,P}(v) := V^{\pi,P}(x_0, v)$. Notice that $V^{\pi,P}(\cdot|r)$ and $V^{\pi,P}(\cdot|g_i)$ encode the value functions for the rewards $r$ and the costs $g_i$ of some constraint $i \in [m]$, respectively. In the following, we sometimes omit the dependency of $V^{\pi,P}(\cdot|v)$ on $P$, when this is clear from context.

## 3.2 OFFLINE OPTIMIZATION IN CMDPs

The goal of the learner in a CMDP is to maximize expected rewards, while at the same time ensuring that all the constraints are satisfied, namely, expected constraint costs are below thresholds. In a CMDP characterized by a reward vector $r \in [0,1]^{|X \times A|}$ and a cost matrix $G \in [0,1]^{|X \times A| \times m}$, such a goal can be formulated by means of the following optimization problem parametrized by $r$ and $G$:

$$\text{OPT}_{r,G} := \begin{cases} \max_{\pi \in \Pi} & V^{\pi}(r) \\ \text{s.t.} & V^{\pi}(g_i) \leq \alpha_i \quad \forall i \in [m]. \end{cases} \tag{1}$$

In this paper, we assume to work with CMDPs that satisfy the following assumption, which is common in the literature on CMDPs (see, *e.g.*, (Efroni et al., 2020)).

**Assumption 1** (Slater's condition). *There exists a strictly feasible policy $\pi^{\diamond} : X \times A \to [0,1]$ such that $V^{\pi}(g_i) < \alpha_i$ for every constraint $i \in [m]$.*

We also need to introduce the Lagrangian function of Problem (1), which is defined as follows.

**Definition 1** (Lagrangian function). *Given a CMDP with reward vector $r \in [0,1]^{|X \times A|}$ and cost matrix $G \in [0,1]^{|X \times A| \times m}$, the Lagrangian function $\mathcal{L}_{r,G} : \Pi \times \mathbb{R}_{\geq 0}^m \to \mathbb{R}$ of Problem (1) is defined as follows, for every policy $\pi \in \Pi$ and vector of Lagrange multipliers $\lambda \in \mathbb{R}_{\geq 0}^m$:*

$$\mathcal{L}_{r,G}(\pi, \lambda) := V^{\pi}(r) - \sum_{i \in [m]} \lambda_i \left(V^{\pi}(g_i) - \alpha_i\right).$$

Finally, we also introduce a problem-specific parameter $\rho \in [0, L]$, strictly related to the feasibility of Problem (1), and in particular to "how much" Slater's condition is satisfied. Formally:

$$\rho := \max_{\pi \in \Pi} \min_{i \in [m]} \left(\alpha_i - V^{\pi}(g_i)\right).$$

The policy leading to the value of $\rho$ is denoted by $\pi^{\circ}$. Intuitively, $\rho$ represents the "margin" by which the "most feasible" strictly feasible policy satisfies the constraints.

## 3.3 ONLINE LEARNING IN EPISODIC CMDPs

We study *episodic* CMDPs in which the interaction in Algorithm 1 is repeated over $T$ episodes. At each episode $t \in [T]$, the learner chooses a policy $\pi_t : X \times A \to [0,1]$, while some reward/cost vectors $r_t \sim \mathcal{R}$ and $g_{t,i} \sim \mathcal{G}_i$ for $i \in [m]$ are sampled by the environment. Then, the interaction goes as in Algorithm 1 with $\pi = \pi_t$, $\widetilde{r} = r_t$, and $\widetilde{g}_i = g_{t,i}$ for all $i \in [m]$. We study the case in which the learner has *bandit feedback*, namely, they only observe the rewards and costs for the state-action pairs visited during the episode. Specifically, at the end of each episode $t \in [T]$, the learner receives as feedback from the environment the sampled rewards $r_t(x_k, a_k)$ and the sampled costs $g_{t,i}(x_k, a_k)$, for every $k \in \{0, \ldots, L-1\}$ and $i \in [m]$. Let us remark that the learner does *not* know anything about reward/cost distributions, as well as the transition function $P$.

In episodic CMDPs, the performance of the learner is measured by means of the two metrics introduced in the following. The *(cumulative) strong regret* over $T$ episodes is defined as:

$$R_T := \sum_{t=1}^{T} \left[\text{OPT}_{r,G} - V^{\pi_t}(r)\right]^{+},$$

where $[\cdot]^+ := \max\{0, \cdot\}$. Intuitively, the regret measures how much the learner loses with respect to always playing an optimal policy. This is a policy solving Problem (1), and it is denoted by $\pi^*$. Thus, since $\mathrm{OPT}_{r,G} = V^{\pi^*}(r)$, we can write the regret as $R_T = \sum_{t=1}^{T} \left[V^{\pi^*}(r) - V^{\pi_t}(r)\right]^+$.

The *(cumulative) strong constraint violation* over $T$ episodes is defined as:

$$V_T := \max_{i \in [m]} \sum_{t=1}^{T} \left[V^{\pi_t}(g_i) - \alpha_i\right]^+.$$

Intuitively, it measures how much the constraints are violated during the learning process.

The goal of the learner is to select policies $\pi_t : X \times A \to [0, 1]$, for each $t \in [T]$, so that both $R_T$ and $V_T$ grow sublinearly in the number of episodes $T$, namely, $R_T = o(T)$ and $V_T = o(T)$.

## 4 PARAMETERS ESTIMATION

Let $N_t(x, a)$ be the number of episodes up to $t \in [T]$ in which the pair $(x, a) \in X \times A$ is *visited*. Then, $\widehat{r}_t(x, a) := \frac{\sum_{\tau \in [t]} r_\tau(x,a) \mathbb{1}_\tau(x,a)}{\max\{1, N_t(x,a)\}}$, with $\mathbb{1}_\tau(x, a) \in \{0, 1\}$ being equal to 1 if and only if $(x, a)$ is visited in episode $\tau$, is an unbiased estimator of the expected reward $r(x, a)$. This immediately follows from the fact that $\widehat{r}_t(x, a)$ is defined as the empirical mean of observed rewards for the state-action pair $(x, a)$. Thus, by applying Hoeffding's inequality, the following lemma holds.

**Lemma 1.** *Given a confidence parameter $\delta \in (0, 1)$, with probability at least $1 - \delta$, the following holds for every episode $t \in [T]$ and state-action pair $(x, a) \in X \times A$:*

$$\left|\widehat{r}_t(x, a) - r(x, a)\right| \leq \phi_t(x, a), \ \text{where } \phi_t(x, a) := \min\left\{1, \sqrt{\frac{4\ln(T|X||A|/\delta)}{\max\{1, N_t(x,a)\}}}\right\}.$$

Similarly, $\widehat{g}_{t,i}(x, a) := \frac{\sum_{\tau \in [t]} g_{\tau,i}(x,a) \mathbb{1}_\tau\{x,a\}}{\max\{1, N_t(x,a)\}}$ is clearly an unbiased estimator of the expected cost $g_i(x, a)$. Thus, by applying Hoeffding's inequality, it is possible to show the following lemma.

**Lemma 2.** *Given a confidence parameter $\delta \in (0, 1)$, with probability at least $1 - \delta$, the following holds for every $i \in [m]$, episode $t \in [T]$, and state-action pair $(x, a) \in X \times A$:*

$$\left|\widehat{g}_{t,i}(x, a) - g_i(x, a)\right| \leq \xi_t(x, a), \ \text{where } \xi_t(x, a) := \min\left\{1, \sqrt{\frac{4\ln(T|X||A|m/\delta)}{\max\{1, N_t(x,a)\}}}\right\}.$$

Moreover, by letting $M_t(x, a, x')$ be the total number of episodes up to $t \in [T]$ in which the state-action pair $(x, a) \in X \times A$ is visited and the environment evolves to the new state $x' \in X$, the estimated transition probability for the triplet $(x, a, x')$ is $\widehat{P}_t(x'|x, a) := \frac{M_t(x,a,x')}{\max\{1, N_t(x,a)\}}$. We refer to Appendix D.2 for additional details and results related to transition probabilities estimation.

**Compact notation** We introduce $\widehat{r}_t \in [0, 1]^{|X \times A|}$ to denote the vector whose components are the estimated rewards $\widehat{r}_t(x, a)$. Moreover, we denote by $\phi_t \in [0, 1]^{|X \times A|}$ the vector whose entries are the bounds $\phi_t(x, a)$. Similarly, we introduce $\widehat{g}_{t,i} \in [0, 1]^{|X \times A|}$ to denote the vector of estimated costs $\widehat{g}_{t,i}(x, a)$, while we denote by $\xi_t \in [0, 1]^{|X \times A|}$ the vector of the bounds $\xi_t(x, a)$. Finally, we let $\overline{r}_t := \widehat{r}_t + \phi_t$, $\underline{r}_t := \widehat{r}_t - \phi_t$, $\overline{g}_{t,i} := \widehat{g}_{t,i} + \xi_t$, and $\underline{g}_{t,i} := \widehat{g}_{t,i} - \xi_t$.

## 5 A NOVEL PRIMAL-DUAL ALGORITHM

Next, we introduce a novel primal-dual algorithm, called *constrained primal-dual policy optimization* (CPD-PO), which allows to efficiently achieve $\widetilde{O}(\sqrt{T})$ *strong* regret and *strong* violation.

### 5.1 THE CPD-PO ALGORITHM

Algorithm 2 shows the pseudocode of CPD-PO. It employs an instance of the *policy optimization with dilated bonus* (PO-DB) algorithm by Luo et al. (2021), which is the state-of-the-art algorithm

---

**Algorithm 2** Constrained Primal-Dual Policy Optimization (`CPD-PO`)

---

**Require:** number of rounds $T \in \mathbb{N}_{>0}$, problem-specific parameter $\rho \in [0, L]$, confidence $\delta \in (0, 1)$
1: $\pi_1 \leftarrow$ first policy prescribed by `PO-DB`
2: Initialize all the estimators, counters, and bounds
3: **for** $t = 1, \ldots, T$ **do**
4:     Interact as in Algorithm 1 with $\pi = \pi_t$, $\widetilde{r} = r_t$, and $\widetilde{g}_i = g_{t,i}$ for $i \in [m]$
5:     Observe $(x_k, a_k)$, $r_t(x_k, a_k)$, and $g_{t,i}(x_k, a_k)$ for every $k \in \{0, \ldots, K-1\}$ and $i \in [m]$, as feedback from the interaction in Algorithm 1
6:     Build estimators $\widehat{r}_t$, $\widehat{g}_{t,i}$, $\phi_t$, $\xi_t$, and $\widehat{P}_t$ as prescribed in Section 4
7:     $\lambda_{t,i} \leftarrow \arg\max_{\lambda \in \{0, \frac{L+1}{\rho}\}} \lambda \left( V^{\pi_t, \widehat{P}_t}(\underline{g}_{t,i}) - \alpha_i \right) \quad \forall i \in [m]$
8:     Build artificial loss for every pair $(x_k, a_k)$ received as feedback:

$$\ell_t(x_k, a_k) \leftarrow \frac{(L+1)m}{\rho} - \left[ \overline{r}_t(x_k, a_k) - \sum_{i \in [m]} \lambda_{t,i} \left( \underline{g}_{t,i}(x_k, a_k) - \frac{\alpha_i}{L} \right) \right]$$

9:     Update policy $\pi_{t+1} \leftarrow$ `PO-DB`$\left( \{(x_k, a_k), \ell_t(x_k, a_k)\}_{k=0}^{L-1} \right)$

---

for learning in adversarial (unconstrained) MDPs under *bandit* feedback. Notice that this algorithm employs a policy optimization approach, by efficiently optimizing state by state. Thus, `PO-DB` does *not* resort to any optimization step performed over the space of occupancy measures.

Algorithm 2 initializes an instance of `PO-DB` as prescribed in (Luo et al., 2021), and it immediately uses it to get the policy $\pi_1$ to be used at $t = 1$. Furthermore, it initializes the estimators $\widehat{r}_t$, $\widehat{g}_{t,i}$ to vectors of zeros and all the counters $N_t(x, a)$, $M_t(x, a, x')$ to zero (Line 2). Then, at every episode $t \in [T]$, the algorithm plays policy $\pi_t$ (Line 4) and observes the feedback associated with the trajectory traversed during the episode (Line 5). The estimators and the confidence bounds are then updated as shown in Section 4 (Line 6). The update of dual variables (Line 7) is performed as a binary choice between two values, namely zero and $^{L+1}/_\rho$, for each $i \in [m]$. The value zero is selected when the optimistic estimation of the $i$-th constraint is *not* violated by the selected policy $\pi_t$ (with respect to the estimated transition $\widehat{P}_t$). In such a case, in the next primal update, the algorithm will *not* focus on minimizing that specific constraint violation. On the contrary, if the optimistic estimation of the $i$-th constraint is *not* satisfied, then the dual update selects the value $^{L+1}/_\rho$. This quantity is chosen to be large enough to guarantee that, with respect to the deterministic Lagrangian game defined by the true reward and cost distributions, any policy cannot gain more than $\text{OPT}_{r,G}$ (see Section 6.1 for further discussion on these aspects). We remark that the value of $V^{\pi_t, \widehat{P}_t}(\underline{g}_{t,i})$ in Line 7 of Algorithm 2 can be efficiently computed by means of a simple dynamic programming procedure. The primal update is performed by the policy update of `PO-DB`. Notice that `PO-DB` is tailored for adversarial MDPs (in which no-statistical assumption is made on the loss functions) with bandit feedback. Thus, for every $t \in [T]$, the primal algorithm expects to receive a loss value for any state-action traversed by the policy previously chosen (and the trajectory itself). We feed `PO-DB` by building a Lagrangian loss, employing the Lagrangian vector selected in Line 7. Notice that the estimated Lagrangian is subtracted and scaled by the state-action maximum (negative) loss $^{(L+1)m}/_\rho$, since `PO-DB` is tailored for positive loss (see Line 8). Finally, the Lagrangian loss and the trajectory are given to the `PO-DB` instance (Line 9). We remark that `PO-DB` minimizes the loss function by employing state-by-state optimization updates. Thus, it is computationally much more efficient than algorithms resorting to a projection on the occupancy measure space.

## 5.2 Algorithm Comparison with (Efroni et al., 2020) and (Müller et al., 2024)

In the following, we highlight the main differences between Algorithm 2 and the primal-dual formulations employed by Efroni et al. (2020) and Müller et al. (2024).

Efroni et al. (2020) were the first to introduce online primal-dual methods that achieve sublinear regret and sublinear constraint violation in CMDPs (allowing for cancellations). Müller et al. (2024) were the first to achieve sublinear *strong* regret and *strong* violation via primal-dual methods.

Efroni et al. (2020) propose two primal-dual algorithms.[3] The first algorithm (`OptDual-CMDP`) performs a UCB-like primal update given an optimistic estimation of the Lagrangian function. Such estimation shares some similarities with ours. As concerns the dual update, `OptDual-CMDP` performs a gradient descent update on the optimistic Lagrangian. The second algorithm (`OptPrimalDual-CMDP`) employs a multiplicative-weight-kind of update for the primal update. Precisely, this update is performed in closed form given the Lagrangified action-value function as objective. The dual update is gradient-descent-like, with the additional modification that the Lagrangian multipliers space is bounded (similarly to our work) since the primal regret minimizer needs the loss/reward functions to be bounded.

Müller et al. (2024) employ similar primal and dual regret minimizers as `OptPrimalDual-CMDP`, namely, multiplicate weights for the primal and gradient descent for the dual. Differently from Efroni et al. (2020), Müller et al. (2024) employ a regularized scheme to define the underlying Lagrangian game played by the primal and the dual regret minimizers. Indeed, the algorithm employs a regularized Lagrangian formulation in order to make the primal objective strictly concave (in the state-action visit distribution) and the dual one strongly convex (in the Lagrangian variable). The strict concavity/strong convexity is necessary to converge in last-iterate to the optimal values of the Lagrangian game and, thus, to attain sublinear *strong* regret and *strong* violation. In our work, we do *not* resort to any regularization scheme, thus simply employing the standard Lagrangian formulation of CMDPs. The main difference between our algorithm and those in (Efroni et al., 2020; Müller et al., 2024) lies in the dual update. Indeed, the black-box primal update employed in our work is multiplicative weight like as the works described above[4]. Differently, in the dual update, we use a UCB-like update, thus not resorting to any adversarial regret minimizers. The UCB-like kind of update is performed between the minimum and the maximum reasonable value for the Lagrange variables. This modification allows us to play the adversarial primal regret minimizer on the deterministic Lagrangian game (up to confidence terms factor) associated with the "best" Lagrangian variable for the violations previously attained. Moreover, notice that our algorithm, since we employ an adversarial primal regret minimizer and differently from algorithms that employ UCB-like updates on the primal (such as `OptDual-CMDP`), may choose non-deterministic policies, which are often optimal in online constrained problem.

# 6 THEORETICAL ANALYSIS

In the following section, we discuss the theoretical guarantees attained by Algorithm 2. Specifically, in Section 6.1 we provide fundamental results on the Lagrangian formulation employed by `CPD-PO`. In Section 6.2 we discuss the guarantees attained by the primal algorithm. Finally, in Section 6.3 we state the regret and violations guarantees attained by Algorithm 2.

## 6.1 RESULTS ON THE LAGRANGIAN FORMULATION

In this section we state some useful preliminary results attained by Algorithm 2. Specifically, the following results concern the primal-dual scheme employed by `CPD-PO`. For the related omitted proof we refer to Appendix C.1. We start by showing that the dual variables decision space is well-defined. Indeed, it is fundamental to show that, given any policy $\pi_t$ selected by the black-box primal algorithm, the dual decision space is sufficient to upper-bound the Lagrangian function value with the constrained optimum. This is done by means of the following lemma.

**Lemma 3.** *Given a CMDP with reward vector $r \in [0,1]^{|X \times A|}$ and cost matrix $G \in [0,1]^{|X \times A| \times m}$, for every policy $\pi \in \Pi$, the following holds:*

$$V^\pi(r) - \max_{\lambda \in \left[0, \frac{L+1}{\rho}\right]^m} \sum_{i \in [m]} \lambda_i \left(V^\pi(g_i) - \alpha_i\right) \leq \text{OPT}_{r,G}.$$

---

[3]It is important to highlight that `OptDual-CMDP` is often referred to as a *dual* method since the primal is only updated given a UCB-like procedure on the Lagrangian. Nevertheless, since the difficulty in attaining *strong* regret and violation holds for dual methods, too, we will refer to this kind of algorithms as primal-dual.

[4]Nevertheless, it is an enhanced version with exploration bonus, which allows to have independent adversarial regret minimizer guarantees (see (Luo et al., 2021) for further discussion)

As previously specified, Lemma 3 states that it is not convenient for the primal algorithm instantiated by Algorithm 2 to play non-safe policies in order to gain more rewards than the one possibly attained by safe policies. Again, this is true given the dual update performed by `CPD-PO`.

It is important to notice that Lemma 3 holds for an optimization problem where rewards, constraints and transitions are known a-priori. Indeed, this is not the case in an online learning setting, where all the aforementioned parameter are unknown and must be estimated in an online fashion. Specifically, if the dual algorithm were aware of the unknown distributions mentioned above, the dual update of Algorithm 2 combined with Lemma 3 would lead to:

$$\mathcal{L}_{r,G}(\pi^*, \lambda_t) \geq \mathcal{L}_{r,G}(\pi_t, \lambda_t), \tag{2}$$

for all $t \in [T]$. Equation (2) would be crucial to prove *strong* regret guarantees, since, it shows that the policy chosen by the algorithm cannot outperform $\pi^*$, thus, making the standard regret definition (on the Lagrangian) to collapse to the *strong* one. Nevertheless, as previously specified, Equation (2) cannot be attained without complete knowledge of the environment. To overcome this issue, we prove the Equation (2) holds *up to* sublinear term, which depends on the uncertainty on the environment estimation. This is done in the following lemma.

**Lemma 4.** *With probability at least $1 - \delta$, Algorithm 2 guarantees that, for every episode $t \in [T]$:*

$$\mathcal{L}_{r,G}(\pi^*, \lambda_t) \geq \mathcal{L}_{r,G}(\pi_t, \lambda_t) - \frac{4Lm}{\rho} V^{\pi_t}(\xi_t) - \frac{8Lm}{\rho} \sum_{x \in X, a \in A} \left| \mathbb{P}_{\pi_t, \widehat{P}_t}(x, a) - \mathbb{P}_{\pi_t, P}(x, a) \right|.$$

The interpretation of Lemma 4 is the following: Equation (2) holds up to two terms. The first term $4Lm/\rho \cdot V^{\pi_t}(\xi_t)$ encompasses the uncertainty on the constraints. Indeed, it is the expectation over policy $\pi_t$ and transitions of the confidence intervals on the costs. This quantity decreases as the algorithm collects cost samples, thus it is sublinear in $T$ when summed over the episodes (we refer to Appendix C.2 for the concentration result). Similarly, the term

$$\frac{8Lm}{\rho} \cdot \sum_{x \in X, a \in A} \left| \mathbb{P}_{\pi_t, \widehat{P}_t}(x, a) - \mathbb{P}_{\pi_t, P}(x, a) \right|,$$

where $\mathbb{P}_{\pi, P}(x, a)$ is the state-action visit distribution given policy $\pi$ and transition $P$, encompasses the uncertainty related to the transitions. This term is sublinear in $T$ when summed over the episode (we refer to Appendix D.2 for the concentration result). Finally, we remark that the uncertainty about the rewards does not affect the dual update, thus, Lemma 4 is completely independent on that.

## 6.2 PRIMAL ALGORITHM

In the following section we state the theoretical guarantees attained by the primal algorithm employed by Algorithm 2. Precisely we have the following guarantees.

**Lemma 5.** *Given any $\delta \in (0, 1)$, the instance of `PO-DB` employed by Algorithm 2 guarantees that, for every policy $\pi \in \Pi$, the primal regret $R_T^{\mathcal{L}}(\pi)$ defined as:*

$$\sum_{t=1}^{T} \left[ V^{\pi}(\overline{r}_t) - \sum_{i \in [m]} \lambda_{t,i} \left( V^{\pi}(\underline{g}_{t,i}) - \alpha_i \right) \right] - \sum_{t=1}^{T} \left[ V^{\pi_t}(\overline{r}_t) - \sum_{i \in [m]} \lambda_{t,i} \left( V^{\pi_t}(\underline{g}_{t,i}) - \alpha_i \right) \right],$$

*is bounded as follows:*

$$R_T^{\mathcal{L}}(\pi) \leq \widetilde{\mathcal{O}} \left( \frac{L^3 m}{\rho} |X| \sqrt{|A|T} + \frac{L^5 m}{\rho} \right),$$

*with probability at least $1 - \mathcal{O}(\delta)$.*

Lemma 5 is obtained by the following considerations. First of all, the analysis employs the no-regret guarantees of `PO-DB` (refer to Appendix D.1 and (Luo et al., 2021) for further discussion). Indeed, by simple computation and using the linearity of expectation, it is possible to recover the $R_T^{\mathcal{L}}(\pi)$ definition by the loss function fed into `PO-DB` by Algorithm 2. Moreover, notice that the no-regret property of `PO-DB` cannot be directly applied to our setting, since the range of the losses is clearly different. Indeed `PO-DB` is tailored for standard episodic MDPs where the losses are bounded in $[0, L]$. This is not true for the Lagrangified MDPs where the losses are bounded in $[0, 2L(L+1)m/\rho]$. Nevertheless, it is sufficient to multiply the original bound for a $\mathcal{O}(Lm/\rho)$ factor to obtain the result.

## 6.3 REGRET AND VIOLATION

In the following section, we provide the cumulative *strong* regret and cumulative *strong* violation guarantees attained by Algorithm 2.

We start showing that a regret of order $\widetilde{\mathcal{O}}(\sqrt{T})$ is attainable by employing primal-dual method which does not optimize over the occupancy measure space. This is done in the following theorem.

**Theorem 1.** *Given any $\delta \in (0,1)$, Algorithm 2 attains:*

$$R_T \leq \widetilde{\mathcal{O}}\left(\frac{L^3 m}{\rho}|X|\sqrt{|A|T} + \frac{L^5 m}{\rho}\right),$$

*with probability at least $1 - \mathcal{O}(\delta)$.*

Theorem 1 is proved by combining Lemma 4 and Lemma 5. Specifically, we start from the theoretical guarantees attained by `PO-DB`, that is, using the bound provided in Lemma 5. Next, employing the optimism of the confidence bound, we recover the true Lagrangian function, excluding sublinear terms. As previously specified, given Lemma 4 it is then possible to recover the cumulative *strong* regret definition on the regret formulated with respect to the Lagrangian function. Finally, to get the final bound, it is sufficient to notice that the optimal solution $\pi^*$ is safe and that:

$$\sum_{i \in [m]} \lambda_{t,i}\left(V^{\pi_t}(g_i) - \alpha_i\right) \geq -\frac{4Lm}{\rho} \sum_{x \in X, a \in A} \left|\mathbb{P}_{\pi_t, \widehat{P}_t}(x,a) - \mathbb{P}_{\pi_t, P}(x,a)\right|.$$

We underline that, from a regret bound perspective, our result obtains the optimal regret order in $T$, while Müller et al. (2024) achieves a suboptimal $T^{0.93}$ only. Furthermore, our bound is not worse than the one in (Müller et al., 2024) w.r.t. the dependency on $\rho, m$ and $L$.

We conclude by stating the result related to the cumulative *strong* violations. Even in this case, we show that the $\widetilde{\mathcal{O}}(\sqrt{T})$ order is attainable.

**Theorem 2.** *For any $\delta \in (0,1)$, Algorithm 2 attains:*

$$V_T \leq \widetilde{\mathcal{O}}\left(L^3 m |X|\sqrt{|A|T} + L^5 m\right),$$

*with probability at least $1 - \mathcal{O}(\delta)$.*

To prove the result, we proceed similarly to Theorem 1, namely, we employ Lemma 5 and we apply the following equality

$$\sum_{i \in [m]} \lambda_{t,i}\left(V^{\pi_t, \widehat{P}_t}(\underline{g}_{t,i}) - \alpha_i\right) = \frac{L}{L+1}\sum_{i \in [m]} \lambda_{t,i}\left(V^{\pi_t, \widehat{P}_t}(\underline{g}_{t,i}) - \alpha_i\right) + \frac{1}{\rho}\sum_{i \in [m]}\left[V^{\pi_t, \widehat{P}_t}(\underline{g}_{t,i}) - \alpha_i\right]^+$$

to retrieve the *strong* violation definition (up to confidence terms). Notice that, given the $L/L+1$ factor which allows us to retrieve the *strong* violation term, it is not possible to directly apply Lemma 4 anymore. Nevertheless, it is indeed possible to prove a similar inequality. Specifically, it holds:

$$\mathcal{L}_{r,G}(\pi^*, \lambda_t) \geq \mathcal{L}_{r,G}(\pi_t, \lambda_t L/L+1) - \frac{2Lm}{\rho}V^{\pi_t}(\xi_t) - \frac{4Lm}{\rho}\sum_{x \in X, a \in A}\left|\mathbb{P}_{\pi_t, \widehat{P}_t}(x,a) - \mathbb{P}_{\pi_t, P}(x,a)\right|.$$

Indeed, the inequality above can be employed to bound sublinearly the following quantity:

$$\sum_{t=1}^{T} \mathcal{L}_{r,G}(\pi_t, \lambda_t L/L+1) - \sum_{t=1}^{T} \mathcal{L}_{r,G}(\pi^*, \lambda_t),$$

which in turn allows to bound sublinearly $\frac{1}{\rho} \cdot \sum_{t=1}^{T} \sum_{i \in [m]} [V^{\pi_t}(g_i) - \alpha_i]^+$, once excluded the sublinear terms associated to the confidence bounds. Finally, simplifying in the `PO-DB` regret bound the $1/\rho$ factor associated to the violation term gives the desired result.

Comparing our violations result with the one in Müller et al. (2024), Algorithm 2 attains the optimal $\widetilde{\mathcal{O}}(\sqrt{T})$ order in $T$, while Müller et al. (2024) achieves a suboptimal $T^{0.93}$. For the violations, our bound is not worse than the one in (Müller et al., 2024) w.r.t. the dependency on $m$ and $L$. Moreover, we do not have the dependency on the Slater's parameter $\rho$.

ACKNOWLEDGMENTS

This paper is supported by the Italian MIUR PRIN 2022 Project "Targeted Learning Dynamics: Computing Efficient and Fair Equilibria through No-Regret Algorithms", by the FAIR (Future Artificial Intelligence Research) project, funded by the NextGenerationEU program within the PNRR-PE-AI scheme (M4C2, Investment 1.3, Line on Artificial Intelligence), and by the EU Horizon project ELIAS (European Lighthouse of AI for Sustainability, No. 101120237).

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

APPENDIX

The appendix is structured as follows:

- In Appendix A, we provide additional related works.

- In Appendix B, we provide the omitted proofs related to the confidence intervals.

- In Appendix C, we provide the omitted proofs of the theoretical guarantees attained by Algorithm 2. Specifically, in Appendix C.1, we provide the results which extend the strong duality ones in constrained Markov decision processes. In Appendix C.2, we provide the results related to the regret bound. In Appendix C.3, we provide the results related to the violations bound.

- In Appendix D, we provide auxiliary technical lemmas from existing works. Precisely, in Appendix D.1, we provide the regret bound of our primal regret minimizer and in Appendix D.2, we further discuss the transition functions confidence intervals.

## A    RELATED WORKS

In the following section, we provide a further discussion on the works which are mainly related to ours. Specifically, we will focus on online learning in unconstrained MDPs, on stochastic online CMDPs and finally on adversarial online CMDPs.

**Online learning in MDPs**    The literature on online learning problems in MDPs is wide. In such settings, two types of feedback are usually studied: in the *full feedback* model, the entire reward/loss function is observed after the learner's choice, while in the *bandit feedback* model, the learner only observes the reward given the chosen action. Azar et al. (2017) study the problem of optimal exploration in episodic MDPs with unknown transitions and stochastic lossee, under bandit feedback. The authors provide an algorithm whose regret upper bound is $\widetilde{\mathcal{O}}(\sqrt{T})$, thus matching the lower bound for this class of MDPs and improving the previous result by Auer et al. (2008). Rosenberg & Mansour (2019b) study the online learning problem in episodic MDPs with adversarial losses and unknown transitions when the feedback is full information. The authors present an online algorithm exploiting entropic regularization and providing a regret upper bound of $\widetilde{\mathcal{O}}(\sqrt{T})$. The same setting is investigated by Rosenberg & Mansour (2019a) when the feedback is bandit. In such a case, the authors provide a regret upper bound of the order of $\widetilde{\mathcal{O}}(T^{3/4})$, which is improved by Jin et al. (2020) by providing an algorithm that achieves in the same setting a regret upper bound of $\widetilde{\mathcal{O}}(\sqrt{T})$. Finally, Luo et al. (2021) provide the first policy optimization algorithm capable of matching the result of (Jin et al., 2020), while avoiding the convex projection on the occupancy measure space.

**Online learning in stochastic CMDPs**    For the stochastic setting, Zheng & Ratliff (2020) deal with episodic CMDPs with stochastic losses and constraints, where the transition probabilities are known and the feedback is bandit. The regret upper bound of their algorithm is of the order of $\widetilde{\mathcal{O}}(T^{3/4})$, while the cumulative constraint violation is guaranteed to be below a threshold with a given probability. Bai et al. (2020) provide the first algorithm that achieves sublinear regret when the transition probabilities are unknown, assuming that the rewards are deterministic and the constraints are stochastic with a particular structure. Efroni et al. (2020) propose two approaches to deal with the exploration-exploitation dilemma in episodic CMDPs. These approaches guarantee sublinear regret and constraint violation when transition probabilities, rewards, and constraints are unknown and stochastic, while the feedback is bandit. Precisely, their LP based methods guarantee $\widetilde{\mathcal{O}}(\sqrt{T})$ cumulative *strong* regret and cumulative *strong* constraints violations. Differently, their primal-dual (or dual) algorithms attain $\widetilde{\mathcal{O}}(\sqrt{T})$ weak regret and violations. Müller et al. (2024) provide the first primal-dual procedure capable of achieving sublinear cumulative *strong* regret and cumulative *strong* constraints violations. Finally, Ghosh et al. (2024) propose a model-free primal-dual algorithm for the linear MDP setting. Their algorithm attains $\widetilde{\mathcal{O}}(\sqrt{T})$ strong violation if it is allowed to take $\Omega(d^{L-1}T^{1.5L}\log(|A|)^L)$ computational steps in every episode. Differently, we focus on polynomial-time algorithms that attain strong regret and violation guarantees.

**Online learning in adversarial CMDPs** In the adversarial setting, Wei et al. (2018) deal with adversarial losses and stochastic constraints, assuming the transition probabilities are known and the feedback is full. The authors provide an algorithm that guarantees an upper bound of the order of $\widetilde{\mathcal{O}}(\sqrt{T})$ on both regret and constraint violation. Qiu et al. (2020) provide a primal-dual approach based on optimism. This work shows the effectiveness of such an approach when dealing with episodic CMDPs with adversarial losses and stochastic constraints, achieving both sublinear regret and constraint violation with full-information feedback. Stradi et al. (2024b) provide the first best-of-both-worlds algorithm for CMDPs, under full feedback. Wei et al. (2023) , Ding & Lavaei (2023) , Stradi et al. (2024c) consider the case in which rewards and constraints are non-stationary, assuming that their variation is bounded. Precisely, all the works mentioned above focus on constraints violation which allows for cancellations. This is not true for Stradi et al. (2024c), where *strong* constraints violation guarantees are provided. Nevertheless, their algorithm is computationally more expensive than a primal-dual procedure, since a linear program must be solved at each episode. Similarly, Stradi et al. (2024a) study the setting with adversarial losses, stochastic constraints and bandit feedback, achieving sublinear regret and sublinear *strong* constraints violations. Again, their algorithm is less efficient than the one we propose in our work, since a convex program must be solved for each episode.

## B    CONFIDENCE INTERVALS

In this section, we report the omitted proof related to the confidence interval employed by our algorithm to deal with the uncertainty on the environments.

**Lemma 6.** *Given any $\delta \in (0,1)$, $i \in [m]$, $t \in [T]$ and $(x,a) \in X \times A$, it holds, with probability at least $1 - \delta$:*

$$\left|\widehat{r}_t(x,a) - r(x,a)\right| \le \iota_t(x,a).$$

*Similarly, with probability at least $1 - \delta$, it holds:*

$$\left|\widehat{g}_{t,i}(x,a) - g_i(x,a)\right| \le \iota_t(x,a),$$

*where $\iota_t(x,a) := \sqrt{\frac{\ln(2/\delta)}{2N_t(x,a)}}$.*

*Proof.* Focus on specifics $t \in [T]$ and $(x,a) \in X \times A$. By Hoeffding's inequality and noticing that rewards values are bounded in $[0,1]$, it holds that:

$$\mathbb{P}\left[\left|\widehat{r}_t(x,a) - r(x,a)\right| \ge \frac{c}{N_t(x,a)}\right] \le 2\exp\left(-\frac{2c^2}{N_t(x,a)}\right)$$

Setting $\delta = 2\exp\left(-\frac{2c^2}{N_t(x,a)}\right)$ and solving to find a proper value of $c$ gives the result for the reward function.

Focusing on specifics $i \in [m]$, $t \in [T]$ and $(x,a) \in X \times A$ and following the previous reasoning applied to the constraints functions concludes the proof. □

### B.1    REWARDS

**Lemma 1.** *Given a confidence parameter $\delta \in (0,1)$, with probability at least $1 - \delta$, the following holds for every episode $t \in [T]$ and state-action pair $(x,a) \in X \times A$:*

$$\left|\widehat{r}_t(x,a) - r(x,a)\right| \le \phi_t(x,a), \ \text{where } \phi_t(x,a) := \min\left\{1, \sqrt{\frac{4\ln(T|X||A|/\delta)}{\max\{1,N_t(x,a)\}}}\right\}.$$

*Proof.* From Lemma 6, given $\delta' \in (0,1)$, we have for any $t \in [T]$ and $(x,a) \in X \times A$:

$$\mathbb{P}\left[\left|\widehat{r}_t(x,a) - r(x,a)\right| \le \iota_t(x,a)\right] \ge 1 - \delta'.$$

Now, we are interested in the intersection of the aforementioned events:

$$\mathbb{P}\left[\bigcap_{x,a,t}\left\{\left|\widehat{r}_t(x,a)-r(x,a)\right|\le\iota_t(x,a)\right\}\right].$$

Thus, we have:

$$\mathbb{P}\left[\bigcap_{x,a,t}\left\{\left|\widehat{r}_t(x,a)-r(x,a)\right|\le\iota_t(x,a)\right\}\right]=1-\mathbb{P}\left[\bigcup_{x,a,t}\left\{\left|\widehat{r}_t(x,a)-r(x,a)\right|\le\iota_t(x,a)\right\}^c\right]$$

$$\ge 1-\sum_{x,a,t}\mathbb{P}\left[\left\{\left|\widehat{r}_t(x,a)-r(x,a)\right|\le\iota_t(x,a)\right\}^c\right]$$

$$\tag{3}$$

$$\ge 1-|X||A|T\delta',$$

where Inequality (3) holds by Union Bound. Noticing that $r_t(x,a)\le 1$, substituting $\delta'$ with $\delta:=\delta'/|X||A|T$ in $\iota_t(x,a)$ with an additional Union Bound over the possible values of $N_t(x,a)$, and thus obtaining $\phi_t(x,a)$, concludes the proof. $\qquad\square$

### B.2  COSTS

**Lemma 2.** *Given a confidence parameter $\delta\in(0,1)$, with probability at least $1-\delta$, the following holds for every $i\in[m]$, episode $t\in[T]$, and state-action pair $(x,a)\in X\times A$:*

$$\left|\widehat{g}_{t,i}(x,a)-g_i(x,a)\right|\le\xi_t(x,a),\ \text{where }\xi_t(x,a):=\min\left\{1,\sqrt{\frac{4\ln(T|X||A|m/\delta)}{\max\{1,N_t(x,a)\}}}\right\}.$$

*Proof.* The proof is analogous to the one of Lemma 1, with an additional Union Bound over the $m$ constraints. $\qquad\square$

## C  PROOFS OMITTED FROM SECTION 6

### C.1  STRONG DUALITY

We start by proving that, when Slater's condition holds (Assumption 1), in an optimal solution the vector of Lagrange multipliers is bounded.

**Lemma 7.** *Given a CMDP with reward vector $r\in[0,1]^{|X\times A|}$ and cost matrix $G\in[0,1]^{|X\times A|\times m}$, it holds:*

$$\min_{\lambda\in\mathbb{R}^m_+:\|\lambda\|_1\in[0,{}^L/\rho]}\max_{\pi\in\Pi}\mathcal{L}_{r,G}(\pi,\lambda)=\max_{\pi\in\Pi}\min_{\lambda\in\mathbb{R}^m_+:\|\lambda\|_1\in[0,{}^L/\rho]}\mathcal{L}_{r,G}(\pi,\lambda)=\text{OPT}_{r,G}.$$

*Proof.* We start by proving the following result:

$$\min_{\lambda\in\mathbb{R}^m_+:\|\lambda\|_1\in[0,{}^L/\rho]}\max_{\pi\in\Pi}\mathcal{L}_{r,G}(\pi,\lambda)=\min_{\lambda\in\mathbb{R}^m_{\ge0}}\max_{\pi\in\Pi}\mathcal{L}_{r,G}(\pi,\lambda).$$

To do so, let us first notice that, for every $\lambda\in\mathbb{R}^m_{\ge0}$ such that $\|\lambda\|_1>{}^L/\rho$:

$$\max_{\pi\in\Pi}\mathcal{L}_{r,G}(\pi,\lambda)\ge\mathcal{L}_{r,G}(\pi^\circ,\lambda)\ge-\sum_{i\in[m]}\lambda_i\left(V^{\pi^\circ}(g_i)-\alpha_i\right)\ge\|\lambda\|_1\rho>L,$$

where we recall that $\pi^\circ:=\arg\max_{\pi\in\Pi}\min_{i\in[m]}\left(\alpha_i-V^\pi(g_i)\right)$. Moreover:

$$\min_{\lambda\in\mathbb{R}^m_+:\|\lambda\|_1\in[0,{}^L/\rho]}\max_{\pi\in\Pi}\mathcal{L}_{r,G}(\pi,\lambda)\le\max_{\pi\in\Pi}\mathcal{L}_{r,G}(\pi,\underline{0})=\max_{\pi\in\Pi}V^\pi(r)\le L.$$

This implies that:

$$\min_{\lambda\in\mathbb{R}^m_{\ge0}}\max_{\pi\in\Pi}\mathcal{L}_{r,G}(\pi,\lambda)=\min\left\{\min_{\lambda\in\mathbb{R}^m_+:\|\lambda\|_1\in[0,{}^L/\rho]}\max_{\pi\in\Pi}\mathcal{L}_{r,G}(\pi,\lambda),\min_{\lambda\in\mathbb{R}^m_+:\|\lambda\|_1>{}^L/\rho}\max_{\pi\in\Pi}\mathcal{L}_{r,G}(\pi,\lambda)\right\}$$

$$= \min_{\lambda \in \mathbb{R}_+^m : \|\lambda\|_1 \in [0, L/\rho]} \max_{\pi \in \Pi} \mathcal{L}_{r,G}(\pi, \lambda).$$

In conclusion,

$$
\begin{aligned}
\mathrm{OPT}_{r,G} &= \max_{\pi \in \Pi} \min_{\lambda \in \mathbb{R}_{\geq 0}^m} \mathcal{L}_{r,G}(\pi, \lambda) \\
&\leq \max_{\pi \in \Pi} \min_{\lambda \in \mathbb{R}_+^m : \|\lambda\|_1 \in [0, L/\rho]} \mathcal{L}_{r,G}(\pi, \lambda) \\
&\leq \min_{\lambda \in \mathbb{R}_+^m : \|\lambda\|_1 \in [0, L/\rho]} \max_{\pi \in \Pi} \mathcal{L}_{r,G}(\pi, \lambda) \\
&= \min_{\lambda \in \mathbb{R}_{\geq 0}^m} \max_{\pi \in \Pi} \mathcal{L}_{r,G}(\pi, \lambda) \\
&= \mathrm{OPT}_{r,G},
\end{aligned}
$$

where the second inequality above holds by the *max-min* inequality and the last equality by strong duality in CMDPs (refer to (Altman, 1999)). This concludes the proof. $\qquad\square$

Now we extend the result showing that, depending on whether an occupancy is safe or not, there exist values of $\lambda$ such that the optimal value of the Lagrangian coincides with the optimum of Program (1).

**Lemma 8.** *For every reward vector $r \in [0, 1]^{|X \times A|}$, constraint cost matrix $G \in [0, 1]^{|X \times A| \times m}$, and threshold vector $\alpha \in [0, L]^m$ and $\pi \in \Pi$ s.t. $V^\pi(g_i) \leq \alpha_i \ \forall i \in [m]$, it holds:*

$$\mathcal{L}_{r,G}(\pi, \underline{0}) \leq OPT_{r,G}.$$

*Proof.* The proof directly follows from the definition of $\mathrm{OPT}_{r,G}$ induced by Program (1). $\qquad\square$

**Lemma 3.** *Given a CMDP with reward vector $r \in [0, 1]^{|X \times A|}$ and cost matrix $G \in [0, 1]^{|X \times A| \times m}$, for every policy $\pi \in \Pi$, the following holds:*

$$V^\pi(r) - \max_{\lambda \in \left[0, \frac{L+1}{\rho}\right]^m} \sum_{i \in [m]} \lambda_i \left(V^\pi(g_i) - \alpha_i\right) \leq \mathrm{OPT}_{r,G}.$$

*Proof.* We analyze separately the case in which $\pi$ satisfies the constraints *(i)* and the case $\pi$ is not safe *(ii)*.

*(i).* We start analyzing $\pi$ s.t. $V^\pi(g_i) \leq \alpha_i$ for all $i \in [m]$. In such a case, it is easy to check that $\arg\max_{\lambda \in [0, \frac{L+1}{\rho}]^m} \sum_{i \in [m]} \lambda_i \left(V^\pi(g_i) - \alpha_i\right) = \underline{0}$; thus, employing Lemma 8 gives the result.

*(ii).* We then consider the case $\pi$ is not safe. In such a case, $\pi$ may either partially satisfy the constraints or violating all of the them. First of all we notice that, in such a scenario, $\arg\max_{\lambda \in [0, \frac{L+1}{\rho}]^m} \sum_{i \in [m]} \lambda_i \left(V^\pi(g_i) - \alpha_i\right) = \bar{\lambda}$ where $\bar{\lambda}$ is the Lagrangian vector composed by $0$ values for the constraints which are violated and $L+1/\rho$ values for the others.

Indeed, it holds:

$$
\begin{aligned}
\mathcal{L}_{r,G}\left(\pi, \bar{\lambda}\right) &= V^\pi(r) - \sum_{i \in [m]} \bar{\lambda}_i \left(V^\pi(g_i) - \alpha_i\right) \\
&\leq \max_{\pi \in \Pi} V^\pi(r) - \frac{L+1}{\rho} \sum_{i \in [m]} \left[V^\pi(g_i) - \alpha_i\right]^+ \quad\quad (4\text{a}) \\
&\leq \max_{\pi \in \Pi} V^\pi(r) - \frac{L}{\rho} \sum_{i \in [m]} \left[V^\pi(g_i) - \alpha_i\right]^+ \\
&\leq \max_{\pi \in \Pi} \min_{\|\lambda\|_1 \in [0, L/\rho]} V^\pi(r) - \sum_{i \in [m]} \lambda_i \left[V^\pi(g_i) - \alpha_i\right]^+ \\
&\leq \min_{\|\lambda\|_1 \in [0, L/\rho]} \max_{\pi \in \Pi} V^\pi(r) - \sum_{i \in [m]} \lambda_i \left[V^\pi(g_i) - \alpha_i\right]^+ \quad\quad (4\text{b})
\end{aligned}
$$

$$\leq \min_{\|\lambda\|_1 \in [0, L/\rho]} \max_{\pi \in \Pi} \mathcal{L}_{r,G}(\pi, \lambda)$$

$$= \text{OPT}_{r,G}, \tag{4c}$$

where the Inequality (4a) holds by definition of $\bar{\lambda}$, the Inequality (4b) holds by the *max-min inequality* and Inequality (4c) follows from Lemma 7. This concludes the proof. □

Finally, we show that the sequence of Lagrangian vector $\lambda_t$ selected by Algorithm 2 guarantees that the value of the Lagrangian attained by $\pi_t$ is always upper bounded by the value of the Lagrangian attained by the optimum $\pi^*$ up to "confidence bound" terms.

**Lemma 4.** *With probability at least $1 - \delta$, Algorithm 2 guarantees that, for every episode $t \in [T]$:*

$$\mathcal{L}_{r,G}(\pi^*, \lambda_t) \geq \mathcal{L}_{r,G}(\pi_t, \lambda_t) - \frac{4Lm}{\rho} V^{\pi_t}(\xi_t) - \frac{8Lm}{\rho} \sum_{x \in X, a \in A} \left| \mathbb{P}_{\pi_t, \widehat{P}_t}(x, a) - \mathbb{P}_{\pi_t, P}(x, a) \right|.$$

*Proof.* First of all, we notice that, when $\lambda_t = \underline{0}$, $\mathcal{L}_{r,G}(\pi^*, \lambda_t) = \text{OPT}_{r,G}$ by definition. Furthermore when $\lambda_t$ is the $\frac{L+1}{\rho}$ vector, it holds that $\mathcal{L}_{r,G}(\pi^*, \lambda_t) \geq \text{OPT}_{r,G}$ since $\pi^*$ is feasible. Same reasoning holds when $\lambda_t$ is any vector in $\left\{0, \frac{L+1}{\rho}\right\}^m$. Thus, it is always the case that $\mathcal{L}_{r,G}(\pi^*, \lambda_t) \geq \text{OPT}_{r,G}$.

Thus, we proceed as follows:

$$\mathcal{L}_{r,G}(\pi_t, \lambda_t)$$

$$\leq V^{\pi_t}(r) - \sum_{i \in [m]} \lambda_{t,i} \left( V^{\pi_t}(\underline{g}_{t,i}) - \alpha_i \right) \tag{5a}$$

$$= V^{\pi_t}(r) - \sum_{i \in [m]} \lambda_{t,i} \left( V^{\pi_t}(\underline{g}_{t,i}) - \alpha_i \right) \pm \sum_{i \in [m]} \lambda_{t,i} \left( V^{\pi_t, \widehat{P}_t}(\underline{g}_{t,i}) - \alpha_i \right)$$

$$\leq V^{\pi_t}(r) - \sum_{i \in [m]} \lambda_{t,i} \left( V^{\pi_t, \widehat{P}_t}(\underline{g}_{t,i}) - \alpha_i \right) + \frac{4Lm}{\rho} \sum_{x \in X, a \in A} \left| \mathbb{P}_{\pi_t, \widehat{P}_t}(x, a) - \mathbb{P}_{\pi_t, P}(x, a) \right| \tag{5b}$$

$$= V^{\pi_t}(r) - \max_{\lambda \in \left\{0, \frac{L+1}{\rho}\right\}^m} \sum_{i \in [m]} \lambda_i \left( V^{\pi_t, \widehat{P}_t}(\underline{g}_{t,i}) - \alpha_i \right)$$

$$+ \frac{4Lm}{\rho} \sum_{x \in X, a \in A} \left| \mathbb{P}_{\pi_t, \widehat{P}_t}(x, a) - \mathbb{P}_{\pi_t, P}(x, a) \right|$$

$$\leq V^{\pi_t}(r) - \max_{\lambda \in \left\{0, \frac{L+1}{\rho}\right\}^m} \sum_{i \in [m]} \lambda_i \left( V^{\pi_t, \widehat{P}_t}(g_i - 2\xi_t) - \alpha_i \right)$$

$$+ \frac{4Lm}{\rho} \sum_{x \in X, a \in A} \left| \mathbb{P}_{\pi_t, \widehat{P}_t}(x, a) - \mathbb{P}_{\pi_t, P}(x, a) \right| \tag{5c}$$

$$= V^{\pi_t}(r) - \max_{\lambda \in \left\{0, \frac{L+1}{\rho}\right\}^m} \sum_{i \in [m]} \lambda_i \left( V^{\pi_t, \widehat{P}_t}(g_i - 2\xi_t) - \alpha_i \right)$$

$$+ \frac{4Lm}{\rho} \sum_{x \in X, a \in A} \left| \mathbb{P}_{\pi_t, \widehat{P}_t}(x, a) - \mathbb{P}_{\pi_t, P}(x, a) \right| \pm \max_{\lambda \in \left\{0, \frac{L+1}{\rho}\right\}^m} \sum_{i \in [m]} \lambda_i \left( V^{\pi_t}(g_i - 2\xi_t) - \alpha_i \right)$$

$$\leq V^{\pi_t}(r) - \max_{\lambda \in \left\{0, \frac{L+1}{\rho}\right\}^m} \sum_{i \in [m]} \lambda_i \left( V^{\pi_t}(g_i - 2\xi_t) - \alpha_i \right)$$

$$+ \frac{8Lm}{\rho} \sum_{x \in X, a \in A} \left| \mathbb{P}_{\pi_t, \widehat{P}_t}(x, a) - \mathbb{P}_{\pi_t, P}(x, a) \right| \tag{5d}$$

$$\leq V^{\pi_t}(r) - \max_{\lambda \in \left\{0, \frac{L+1}{\rho}\right\}^m} \sum_{i \in [m]} \lambda_i \left( V^{\pi_t}(g_i) - \alpha_i \right) + \frac{4Lm}{\rho} V^{\pi_t}(\xi_t)$$

$$+ \frac{8Lm}{\rho} \sum_{x \in X, a \in A} \left| \mathbb{P}_{\pi_t, \widehat{P}_t}(x, a) - \mathbb{P}_{\pi_t, P}(x, a) \right| \tag{5e}$$

$$= V^{\pi_t}(r) - \max_{\lambda \in [0, \frac{L+1}{\rho}]^m} \sum_{i \in [m]} \lambda_i \left(V^{\pi_t}(g_i) - \alpha_i\right) + \frac{4Lm}{\rho} V^{\pi_t}(\xi_t)$$

$$+ \frac{8Lm}{\rho} \sum_{x \in X, a \in A} \left| \mathbb{P}_{\pi_t, \widehat{P}_t}(x, a) - \mathbb{P}_{\pi_t, P}(x, a) \right|$$

$$\leq \mathrm{OPT}_{r, G} + \frac{4Lm}{\rho} V^{\pi_t}(\xi_t) + \frac{8Lm}{\rho} \sum_{x \in X, a \in A} \left| \mathbb{P}_{\pi_t, \widehat{P}_t}(x, a) - \mathbb{P}_{\pi_t, P}(x, a) \right|, \qquad (5f)$$

where Inequality (5a) holds with probability at least $1 - \delta$ by Lemma 2, Inequality (5b) holds by Hölder inequality, Inequality (5c) holds with probability at least $1 - \delta$ by Lemma 2, Inequality (5d) holds by Hölder inequality, Inequality (5e) holds by definition of $\lambda_t$ and Inequality (5f) holds by Lemma 3.

Thus, it holds:

$$\mathcal{L}_{r, G}(\pi^*, \lambda_t) \geq \mathrm{OPT}_{r, G}$$

$$\geq \mathcal{L}_{r, G}(\pi_t, \lambda_t) - \frac{4Lm}{\rho} V^{\pi_t}(\xi_t) - \frac{8Lm}{\rho} \sum_{x \in X, a \in A} \left| \mathbb{P}_{\pi_t, \widehat{P}_t}(x, a) - \mathbb{P}_{\pi_t, P}(x, a) \right|,$$

which concludes the proof. $\qquad \square$

**Lemma 9.** *With probability at least $1 - \delta$, Algorithm 2 guarantees, for all $t \in [T]$:*

$$\mathcal{L}_{r, G}(\pi^*, \lambda_t) \geq \mathcal{L}_{r, G}(\pi_t, \lambda_t L / L + 1) - \frac{2Lm}{\rho} V^{\pi_t}(\xi_t) - \frac{4Lm}{\rho} \sum_{x \in X, a \in A} \left| \mathbb{P}_{\pi_t, \widehat{P}_t}(x, a) - \mathbb{P}_{\pi_t, P}(x, a) \right|.$$

*Proof.* Similarly to Lemma 4, we notice that $\mathcal{L}_{r, G}(\pi^*, \lambda_t) \geq \mathrm{OPT}_{r, G}$.

Thus, following the same steps as in Lemma 4, we proceed as follows:

$$\mathcal{L}_{r, G}(\pi_t, \lambda_t L / L + 1)$$

$$\leq V^{\pi_t}(r) - \max_{\lambda \in \{0, \frac{L+1}{\rho}\}^m} \sum_{i \in [m]} \frac{\lambda_i L}{L + 1} \left(V^{\pi_t}(g_i) - \alpha_i\right) + \frac{2Lm}{\rho} V^{\pi_t}(\xi_t)$$

$$+ \frac{4Lm}{\rho} \sum_{x \in X, a \in A} \left| \mathbb{P}_{\pi_t, \widehat{P}_t}(x, a) - \mathbb{P}_{\pi_t, P}(x, a) \right|$$

$$= V^{\pi_t}(r) - \max_{\lambda \in [0, \frac{L+1}{\rho}]^m} \sum_{i \in [m]} \frac{\lambda_i L}{L + 1} \left(V^{\pi_t}(g_i) - \alpha_i\right) + \frac{2Lm}{\rho} V^{\pi_t}(\xi_t)$$

$$+ \frac{4Lm}{\rho} \sum_{x \in X, a \in A} \left| \mathbb{P}_{\pi_t, \widehat{P}_t}(x, a) - \mathbb{P}_{\pi_t, P}(x, a) \right|$$

$$= V^{\pi_t}(r) - \max_{\lambda \in [0, \frac{L}{\rho}]^m} \sum_{\in [m]} \lambda_i \left(V^{\pi_t}(g_i) - \alpha_i\right) + \frac{2Lm}{\rho} V^{\pi_t}(\xi_t)$$

$$+ \frac{4Lm}{\rho} \sum_{x \in X, a \in A} \left| \mathbb{P}_{\pi_t, \widehat{P}_t}(x, a) - \mathbb{P}_{\pi_t, P}(x, a) \right|$$

$$\leq \max_{\pi \in \Pi} \left[ V^{\pi}(r) - \max_{\lambda \in [0, \frac{L}{\rho}]^m} \sum_{i \in [m]} \lambda_i \left(V^{\pi}(g_i) - \alpha_i\right) \right] + \frac{2Lm}{\rho} V^{\pi_t}(\xi_t)$$

$$+ \frac{4Lm}{\rho} \sum_{x \in X, a \in A} \left| \mathbb{P}_{\pi_t, \widehat{P}_t}(x, a) - \mathbb{P}_{\pi_t, P}(x, a) \right|$$

$$\leq \max_{\pi \in \Pi} \left[ V^{\pi}(r) - \max_{\|\lambda\|_1 \in [0, \frac{L}{\rho}]} \sum_{i \in [m]} \lambda_i \left(V^{\pi}(g_i) - \alpha_i\right) \right] + \frac{2Lm}{\rho} V^{\pi_t}(\xi_t)$$

$$+ \frac{4Lm}{\rho} \sum_{x \in X, a \in A} \left| \mathbb{P}_{\pi_t, \widehat{P}_t}(x, a) - \mathbb{P}_{\pi_t, P}(x, a) \right|$$

$$\leq \text{OPT}_{r,G} + \frac{2Lm}{\rho} V^{\pi_t}(\xi_t) + \frac{4Lm}{\rho} \sum_{x \in X, a \in A} \left| \mathbb{P}_{\pi_t, \widehat{P}_t}(x, a) - \mathbb{P}_{\pi_t, P}(x, a) \right|, \tag{6}$$

where Inequality (6) holds by Lemma 7.

Thus, it holds:

$$\mathcal{L}_{r,G}(\pi^*, \lambda_t) \geq \text{OPT}_{r,G}$$

$$\geq \mathcal{L}_{r,G}(\pi_t, \lambda_t L / L + 1) - \frac{2Lm}{\rho} V^{\pi_t}(\xi_t) - \frac{4Lm}{\rho} \sum_{x \in X, a \in A} \left| \mathbb{P}_{\pi_t, \widehat{P}_t}(x, a) - \mathbb{P}_{\pi_t, P}(x, a) \right|,$$

which concludes the proof. $\qquad \square$

## C.2 REGRET

We first show the regret bound attained by PO-DB on the Lagrangian loss built by Algorithm 2. Precisely, it holds the following result.

**Lemma 5.** *Given any $\delta \in (0, 1)$, the instance of* PO-DB *employed by Algorithm 2 guarantees that, for every policy $\pi \in \Pi$, the primal regret $R_T^{\mathcal{L}}(\pi)$ defined as:*

$$\sum_{t=1}^{T} \left[ V^{\pi}(\overline{r}_t) - \sum_{i \in [m]} \lambda_{t,i} \left( V^{\pi}(\underline{g}_{t,i}) - \alpha_i \right) \right] - \sum_{t=1}^{T} \left[ V^{\pi_t}(\overline{r}_t) - \sum_{i \in [m]} \lambda_{t,i} \left( V^{\pi_t}(\underline{g}_{t,i}) - \alpha_i \right) \right],$$

*is bounded as follows:*

$$R_T^{\mathcal{L}}(\pi) \leq \widetilde{\mathcal{O}} \left( \frac{L^3 m}{\rho} |X| \sqrt{|A|T} + \frac{L^5 m}{\rho} \right),$$

*with probability at least $1 - \mathcal{O}(\delta)$.*

*Proof.* We first notice that, by simple computation, it holds:

$$\sum_{t=1}^{T} V^{\pi_t} \left( \frac{(L+1)m}{\rho} \right) - \sum_{t=1}^{T} \left[ V^{\pi_t}(\overline{r}_t) - \sum_{i \in [m]} \lambda_{t,i} V^{\pi_t} \left( \underline{g}_{t,i} - \frac{\alpha_i}{L} \right) \right]$$

$$- \sum_{t=1}^{T} V^{\pi} \left( \frac{(L+1)m}{\rho} \right) - \sum_{t=1}^{T} \left[ V^{\pi}(\overline{r}_t) - \sum_{i \in [m]} \lambda_{t,i} V^{\pi} \left( \underline{g}_{t,i} - \frac{\alpha_i}{L} \right) \right]$$

$$= - \sum_{t=1}^{T} \left[ V^{\pi_t}(\overline{r}_t) - \sum_{i \in [m]} \lambda_{t,i} V^{\pi_t} \left( \underline{g}_{t,i} - \frac{\alpha_i}{L} \right) \right] + \sum_{t=1}^{T} \left[ V^{\pi}(\overline{r}_t) - \sum_{i \in [m]} \lambda_{t,i} V^{\pi} \left( \underline{g}_{t,i} - \frac{\alpha_i}{L} \right) \right]$$

$$= \sum_{t=1}^{T} \left[ V^{\pi}(\overline{r}_t) - \sum_{i \in [m]} \lambda_{t,i} \left( V^{\pi}(\underline{g}_{t,i}) - \alpha_i \right) \right] - \sum_{t=1}^{T} \left[ V^{\pi_t}(\overline{r}_t) - \sum_{i \in [m]} \lambda_{t,i} \left( V^{\pi_t}(\underline{g}_{t,i}) - \alpha_i \right) \right],$$

where the first step holds since $(L+1)m/\rho$ is constant for any state-action pair and the second step holds since $V^{\pi}(\alpha_i/L) = \alpha_i$.

Then, we notice that, any regret minimizer algorithm $\mathcal{A}$, which attains $\mathcal{E}_{\mathcal{A}}$ regret upper-bound when the losses are bounded in $[0, 1]$ (or in $[0, L]$ for MDPs), may achieve $C \cdot \mathcal{E}_{\mathcal{A}}$ regret upper-bound when the range of the losses is $C$ and it is known to the algorithm. The result is intuitive, since it is always possible to apply an affine transformation to the losses received and, thus, to scale them in $[0, 1]$. Then, the algorithm can be fed with the scaled loss as expected, but it will suffer a loss multiplied by a $C$ factor, attaining the aforementioned $C \cdot \mathcal{E}_{\mathcal{A}}$ bound.

Noticing that the loss vector received by PO-DB is multiplied by an additional $\mathcal{O}(Lm/\rho)$ factor given by the definition of the Lagrangian variable in Algorithm 2, we can employ the standard regret

bound PO-DB (see Lemma 12) with the additional $\mathcal{O}(Lm/\rho)$ factor. Thus, it holds, with probability at least $1 - \mathcal{O}(\delta)$:

$$
\sum_{t=1}^{T} \left[ V^{\pi}(\overline{r}_t) - \sum_{i \in [m]} \lambda_{t,i} \left( V^{\pi}(\underline{g}_{t,i}) - \alpha_i \right) \right] - \sum_{t=1}^{T} \left[ V^{\pi_t}(\overline{r}_t) - \sum_{i \in [m]} \lambda_{t,i} \left( V^{\pi_t}(\underline{g}_{t,i}) - \alpha_i \right) \right]
$$

$$
= \sum_{t=1}^{T} V^{\pi_t} \left( \frac{(L+1)m}{\rho} \right) - \sum_{t=1}^{T} \left[ V^{\pi_t}(\overline{r}_t) - \sum_{i \in [m]} \lambda_{t,i} V^{\pi_t} \left( \underline{g}_{t,i} - \frac{\alpha_i}{L} \right) \right]
$$

$$
- \sum_{t=1}^{T} V^{\pi} \left( \frac{(L+1)m}{\rho} \right) - \sum_{t=1}^{T} \left[ V^{\pi}(\overline{r}_t) - \sum_{i \in [m]} \lambda_{t,i} V^{\pi} \left( \underline{g}_{t,i} - \frac{\alpha_i}{L} \right) \right]
$$

$$
\leq \widetilde{\mathcal{O}} \left( \frac{L^3 m}{\rho} |X| \sqrt{|A|T} + \frac{L^5 m}{\rho} \right),
$$

which concludes the proof. $\qquad\square$

Then, we show the concentration rate of the confidence intervals.

**Lemma 10.** *With probability at least $1 - \delta$, it holds:*

$$
\sum_{t=1}^{T} V^{\pi_t}(\phi_t) \leq 4\sqrt{L|X||A|T \ln \left( \frac{T|X||A|}{\delta} \right)} + L\sqrt{2T \ln \frac{1}{\delta}}
$$

*Proof.* We first notice the following bound,

$$
V^{\pi_t}(\phi_t) \leq L.
$$

Thus, we can employ the Azuma inequality to bound the following Martingale difference sequence as

$$
\sum_{t=1}^{T} V^{\pi_t}(\phi_t) - \sum_{t=1}^{T} \sum_{x,a} \phi_t(x,a) \mathbb{1}_t\{x,a\} \leq L\sqrt{2T \ln \frac{1}{\delta}},
$$

which holds with probability $1 - \delta$. Thus we can bound the quantity of interest as follows,

$$
\sum_{t=1}^{T} V^{\pi_t}(\phi_t) \leq \sum_{t=1}^{T} \sum_{x,a} \phi_t(x,a) \mathbb{1}_t\{x,a\} + L\sqrt{2T \ln \frac{1}{\delta}}
$$

$$
= \sqrt{4 \ln \left( \frac{T|X||A|}{\delta} \right)} \sum_{t=1}^{T} \sum_{x,a} \sqrt{\frac{1}{\max\{1, N_t(x,a)\}}} \mathbb{1}_t\{x,a\} + L\sqrt{2T \ln \frac{1}{\delta}}
$$

$$
\leq 2\sqrt{4 \ln \left( \frac{T|X||A|}{\delta} \right)} \sum_{x,a} \sqrt{N_T(x,a)} + L\sqrt{2T \ln \frac{1}{\delta}} \tag{7}
$$

$$
\leq 4\sqrt{L|X||A|T \ln \left( \frac{T|X||A|}{\delta} \right)} + L\sqrt{2T \ln \frac{1}{\delta}}, \tag{8}
$$

where Inequality (7) holds since $\sum_{t=1}^{T} \frac{1}{t} \leq 2\sqrt{T}$ and Inequality (8) follows from Cauchy-Schwarz inequality and noticing that $\sqrt{\sum_{x,a} N_T(x,a)} \leq \sqrt{LT}$. This concludes the proof. $\qquad\square$

**Lemma 11.** *With probability at least $1 - \delta$, it holds:*

$$
\sum_{t=1}^{T} V^{\pi_t}(\xi_t) \leq 4\sqrt{L|X||A|T \ln \left( \frac{T|X||A|m}{\delta} \right)} + L\sqrt{2T \ln \frac{1}{\delta}}
$$

*Proof.* The proof follows the one of Lemma 10, replacing $\phi_t$ with $\xi_t$. $\qquad\square$

We are now ready to prove the regret bound of Algorithm 2.

**Theorem 1.** *Given any $\delta \in (0,1)$, Algorithm 2 attains:*

$$R_T \leq \widetilde{\mathcal{O}} \left( \frac{L^3 m}{\rho} |X| \sqrt{|A|T} + \frac{L^5 m}{\rho} \right),$$

*with probability at least $1 - \mathcal{O}(\delta)$.*

*Proof.* We first notice that, by Lemma 4, we have that:

$$\mathcal{L}_{r,G}(\pi^*, \lambda_t) \geq \mathcal{L}_{r,G}(\pi_t, \lambda_t) - \frac{4Lm}{\rho} V^{\pi_t}(\xi_t) - \frac{8Lm}{\rho} \sum_{x \in X, a \in A} \left| \mathbb{P}_{\pi_t, \widehat{P}_t}(x,a) - \mathbb{P}_{\pi_t, P}(x,a) \right|, \quad \forall t \in [T],$$

with probability at least $1 - \delta$ and where $\pi^*$ is the optimal solution corresponding to $\mathrm{OPT}_{r,G}$. Thus, we have that an upper bound on:

$$\sum_{t=1}^T \left( \mathcal{L}_{r,G}(\pi^*, \lambda_t) - \mathcal{L}_{r,G}(\pi_t, \lambda_t) + \frac{4Lm}{\rho} V^{\pi_t}(\xi_t) + \frac{8Lm}{\rho} \sum_{x \in X, a \in A} \left| \mathbb{P}_{\pi_t, \widehat{P}_t}(x,a) - \mathbb{P}_{\pi_t, P}(x,a) \right| \right) \tag{9}$$

enforces the same bound on

$$\sum_{t=1}^T \left[ \mathcal{L}_{r,G}(\pi^*, \lambda_t) - \mathcal{L}_{r,G}(\pi_t, \lambda_t) + \frac{4Lm}{\rho} V^{\pi_t}(\xi_t) + \frac{8Lm}{\rho} \sum_{x \in X, a \in A} \left| \mathbb{P}_{\pi_t, \widehat{P}_t}(x,a) - \mathbb{P}_{\pi_t, P}(x,a) \right| \right]^+,$$

since the term in the summation is always non-negative.

Now, we employ Lemma 5, which implies that with probability at least $1 - \mathcal{O}(\delta)$, for any $\pi \in \Pi$:

$$\sum_{t=1}^T \left( V^\pi(\overline{r}_t) - \sum_{i \in [m]} \lambda_{t,i} \left( V^\pi(\underline{g}_{t,i}) - \alpha_i \right) \right) - \sum_{t=1}^T \left( V^{\pi_t}(\overline{r}_t) - \sum_{i \in [m]} \lambda_{t,i} \left( V^{\pi_t}(\underline{g}_{t,i}) - \alpha_i \right) \right)$$
$$\leq \widetilde{\mathcal{O}} \left( \frac{L^3 m}{\rho} |X| \sqrt{|A|T} + \frac{L^5 m}{\rho} \right), \tag{10}$$

which implies the same bound for $\pi^*$.

We now show that the left-hand side of the aforementioned equation reduce to Equation (9) up to sublinear terms. By Lemma 1, with probability at least $1 - \delta$, it holds:

$$r \preceq \overline{r}_t.$$

Thus, taking the expectation over policy and transition we obtain:

$$V^\pi(r) \leq V^\pi(\overline{r}_t).$$

Thus, Equation (10) can be rewritten as:

$$\sum_{t=1}^T \left( V^{\pi^*}(r) - \sum_{i \in [m]} \lambda_{t,i} \left( V^{\pi^*}(\underline{g}_{t,i}) - \alpha_i \right) \right) - \sum_{t=1}^T \left( V^{\pi_t}(\overline{r}_t) - \sum_{i \in [m]} \lambda_{t,i} \left( V^{\pi_t}(\underline{g}_{t,i}) - \alpha_i \right) \right)$$
$$\leq \widetilde{\mathcal{O}} \left( \frac{L^3 m}{\rho} |X| \sqrt{|A|T} + \frac{L^5 m}{\rho} \right).$$

Similarly, it holds that $\widehat{r}_t \preceq r + \phi_t$ which implies:

$$\sum_{t=1}^T \left( V^{\pi^*}(r) - \sum_{i \in [m]} \lambda_{t,i} \left( V^{\pi^*}(\underline{g}_{t,i}) - \alpha_i \right) \right) - \sum_{t=1}^T \left( V^{\pi_t}(r) - \sum_{i \in [m]} \lambda_{t,i} \left( V^{\pi_t}(\underline{g}_{t,i}) - \alpha_i \right) \right)$$
$$\leq \widetilde{\mathcal{O}} \left( \frac{L^3 m}{\rho} |X| \sqrt{|A|T} + \frac{L^5 m}{\rho} \right) + 2 \sum_{t=1}^T V^{\pi_t}(\phi_t).$$

We apply Lemma 10 to obtain, with probability at least $1 - \mathcal{O}(\delta)$, by Union Bound:

$$\sum_{t=1}^{T} \left( V^{\pi^*}(r) - \sum_{i \in [m]} \lambda_{t,i} \left( V^{\pi^*}(\underline{g}_{t,i}) - \alpha_i \right) \right) - \sum_{t=1}^{T} \left( V^{\pi_t}(r) - \sum_{i \in [m]} \lambda_{t,i} \left( V^{\pi_t}\left( \underline{g}_{t,i} \right) - \alpha_i \right) \right)$$
$$\leq \widetilde{\mathcal{O}} \left( \frac{L^3 m}{\rho} |X| \sqrt{|A|T} + \frac{L^5 m}{\rho} \right).$$

We proceed similarly for the constraints cost noticing that $\underline{g}_{t,i} \preceq g_i$, and $g_i - 2\xi_t \preceq \underline{g}_{t,i}$ for all $i \in [m]$. Thus, we have:

$$\sum_{t=1}^{T} \left( V^{\pi^*}(r) - \sum_{i \in [m]} \lambda_{t,i} \left( V^{\pi^*}(g_i) - \alpha_i \right) \right) - \sum_{t=1}^{T} \left( V^{\pi_t}(r) - \sum_{i \in [m]} \lambda_{t,i} \left( V^{\pi_t}(g_i) - \alpha_i \right) \right)$$
$$\leq \widetilde{\mathcal{O}} \left( \frac{L^3 m}{\rho} |X| \sqrt{|A|T} + \frac{L^5 m}{\rho} \right) + 2m \sum_{t=1}^{T} V^{\pi_t}(\xi_t).$$

Finally, employing Lemma 11, we have that the regret defined over the Lagrangian is bounded by,

$$\sum_{t=1}^{T} \left( V^{\pi^*}(r) - \sum_{i \in [m]} \lambda_{t,i} \left( V^{\pi^*}(g_i) - \alpha_i \right) \right) - \sum_{t=1}^{T} \left( V^{\pi_t}(r) - \sum_{i \in [m]} \lambda_{t,i} \left( V^{\pi_t}(g_i) - \alpha_i \right) \right)$$
$$\leq \widetilde{\mathcal{O}} \left( \frac{L^3 m}{\rho} |X| \sqrt{|A|T} + \frac{L^5 m}{\rho} \right).$$

Now, we notice the following inequality,

$$\sum_{t=1}^{T} \left( \left( V^{\pi^*}(r) - \sum_{i \in [m]} \lambda_{t,i} \left( V^{\pi^*}(g_i) - \alpha_i \right) \right) - \left( V^{\pi_t}(r) - \sum_{i \in [m]} \lambda_{t,i} \left( V^{\pi_t}(g_i) - \alpha_i \right) \right) \right.$$
$$\left. \pm \left( \frac{4Lm}{\rho} V^{\pi_t}(\xi_t) + \frac{8Lm}{\rho} \sum_{x \in X, a \in A} \left| \mathbb{P}_{\pi_t, \widehat{P}_t}(x, a) - \mathbb{P}_{\pi_t, P}(x, a) \right| \right) \right)$$
$$\leq \widetilde{\mathcal{O}} \left( \frac{L^3 m}{\rho} |X| \sqrt{|A|T} + \frac{L^5 m}{\rho} \right),$$

which implies the following chain of inequalities,

$$\sum_{t=1}^{T} \left( \left( V^{\pi^*}(r) - \sum_{i \in [m]} \lambda_{t,i} \left( V^{\pi^*}(g_i) - \alpha_i \right) \right) - \left( V^{\pi_t}(r) - \sum_{i \in [m]} \lambda_{t,i} \left( V^{\pi_t}(g_i) - \alpha_i \right) \right) \right.$$
$$\left. + \frac{4Lm}{\rho} V^{\pi_t}(\xi_t) + \frac{8Lm}{\rho} \sum_{x \in X, a \in A} \left| \mathbb{P}_{\pi_t, \widehat{P}_t}(x, a) - \mathbb{P}_{\pi_t, P}(x, a) \right| \right)$$
$$\leq \widetilde{\mathcal{O}} \left( \frac{L^3 m}{\rho} |X| \sqrt{|A|T} + \frac{L^5 m}{\rho} \right) + \sum_{t=1}^{T} \frac{4Lm}{\rho} V^{\pi_t}(\xi_t)$$
$$+ \sum_{t=1}^{T} \frac{8Lm}{\rho} \sum_{x \in X, a \in A} \left| \mathbb{P}_{\pi_t, \widehat{P}_t}(x, a) - \mathbb{P}_{\pi_t, P}(x, a) \right|$$
$$\leq \widetilde{\mathcal{O}} \left( \frac{L^3 m}{\rho} |X| \sqrt{|A|T} + \frac{L^5 m}{\rho} \right),$$

where in the last step we employed Lemma 11 and Lemma 14.

We then employ Lemma 4 to get the following bound:

$$\sum_{t=1}^{T} \left[ \mathcal{L}_{r,G}(\pi^*, \lambda_t) - \mathcal{L}_{r,G}(\pi_t, \lambda_t) + \frac{4Lm}{\rho} V^{\pi_t}(\xi_t) + \frac{8Lm}{\rho} \sum_{x \in X, a \in A} \left| \mathbb{P}_{\pi_t, \widehat{P}_t}(x, a) - \mathbb{P}_{\pi_t, P}(x, a) \right| \right]^+$$

$$\leq \widetilde{\mathcal{O}}\left(\frac{L^3 m}{\rho}|X|\sqrt{|A|T} + \frac{L^5 m}{\rho}\right),$$

which, since $\frac{4Lm}{\rho}V^{\pi_t}(\xi_t)$ and $\frac{8Lm}{\rho}\left|\mathbb{P}_{\pi_t,\widehat{P}_t}(x,a) - \mathbb{P}_{\pi_t,P}(x,a)\right|$ are non-negative by construction, implies:

$$\sum_{t=1}^{T}\left[\left(V^{\pi^*}(r) - \sum_{i\in[m]}\lambda_{t,i}\left(V^{\pi^*}(g_i) - \alpha_i\right)\right) - \left(V^{\pi_t}(r) - \sum_{i\in[m]}\lambda_{t,i}\left(V^{\pi_t}(g_i) - \alpha_i\right)\right)\right]^+$$
$$\leq \widetilde{\mathcal{O}}\left(\frac{L^3 m}{\rho}|X|\sqrt{|A|T} + \frac{L^5 m}{\rho}\right).$$

To get the final bound, we first notice that, by definition of the constrained optimum $\pi^*$, the quantity $V^{\pi^*}(g_i) - \alpha_i$ is always non-positive for every $i \in [m]$. Moreover, by the Lagrangian update of Algorithm 2, we have that:

$$\sum_{i\in[m]}\lambda_{t,i}\left(V^{\pi_t}(g_i) - \alpha_i\right)$$
$$\geq \sum_{i\in[m]}\lambda_{t,i}\left(V^{\pi_t}(\underline{g}_{t,i}) - \alpha_i\right)$$
$$= \sum_{i\in[m]}\lambda_{t,i}\left(V^{\pi_t}(\underline{g}_{t,i}) - \alpha_i\right) \pm \sum_{i\in[m]}\lambda_{t,i}\left(V^{\pi_t,\widehat{P}_t}(\underline{g}_{t,i}) - \alpha_i\right)$$
$$\geq \sum_{i\in[m]}\lambda_{t,i}\left(V^{\pi_t,\widehat{P}_t}(\underline{g}_{t,i}) - \alpha_i\right) - \frac{4Lm}{\rho}\sum_{x\in X, a\in A}\left|\mathbb{P}_{\pi_t,\widehat{P}_t}(x,a) - \mathbb{P}_{\pi_t,P}(x,a)\right|$$
$$\geq -\frac{4Lm}{\rho}\sum_{x\in X, a\in A}\left|\mathbb{P}_{\pi_t,\widehat{P}_t}(x,a) - \mathbb{P}_{\pi_t,P}(x,a)\right|.$$

Thus, following the reasoning above, we bound the positive cumulative regret as follows:

$$R_T := \sum_{t=1}^{T}\left[V^{\pi^*}(r) - V^{\pi_t}(r)\right]^+$$
$$\leq \sum_{t=1}^{T}\left[\left(V^{\pi^*}(r) - \sum_{i\in[m]}\lambda_{t,i}\left(V^{\pi^*}(g_i) - \alpha_i\right)\right) - \left(V^{\pi_t}(r) - \sum_{i\in[m]}\lambda_{t,i}\left(V^{\pi_t}(g_i) - \alpha_i\right)\right)\right.$$
$$\left. + \frac{4Lm}{\rho}\sum_{x\in X, a\in A}\left|\mathbb{P}_{\pi_t,\widehat{P}_t}(x,a) - \mathbb{P}_{\pi_t,P}(x,a)\right|\right]^+$$
$$\leq \sum_{t=1}^{T}\left[\left(V^{\pi^*}(r) - \sum_{i\in[m]}\lambda_{t,i}\left(V^{\pi^*}(g_i) - \alpha_i\right)\right) - \left(V^{\pi_t}(r) - \sum_{i\in[m]}\lambda_{t,i}\left(V^{\pi_t}(g_i) - \alpha_i\right)\right)\right]^+$$
$$+ \sum_{t=1}^{T}\left[\frac{4Lm}{\rho}\sum_{x\in X, a\in A}\left|\mathbb{P}_{\pi_t,\widehat{P}_t}(x,a) - \mathbb{P}_{\pi_t,P}(x,a)\right|\right]^+$$
$$\leq \widetilde{\mathcal{O}}\left(\frac{L^3 m}{\rho}|X|\sqrt{|A|T} + \frac{L^5 m}{\rho}\right) + \sum_{t=1}^{T}\frac{4Lm}{\rho}\sum_{x\in X, a\in A}\left|\mathbb{P}_{\pi_t,\widehat{P}_t}(x,a) - \mathbb{P}_{\pi_t,P}(x,a)\right|$$
$$\leq \widetilde{\mathcal{O}}\left(\frac{L^3 m}{\rho}|X|\sqrt{|A|T} + \frac{L^5 m}{\rho}\right),$$

which holds with probability $1 - \mathcal{O}(\delta)$ by Lemma 14 and Union Bound. This concludes the proof. $\qquad\square$

## C.3 Violations

**Theorem 2.** *For any $\delta \in (0,1)$, Algorithm 2 attains:*

$$V_T \leq \widetilde{\mathcal{O}}\left(L^3 m |X| \sqrt{|A|T} + L^5 m\right),$$

*with probability at least $1 - \mathcal{O}(\delta)$.*

*Proof.* We first notice the following equations:

$$\sum_{i \in [m]} \lambda_{t,i} \left(V^{\pi_t, \widehat{P}_t}(\underline{g}_{t,i}) - \alpha_i\right)$$

$$= \frac{L}{L+1} \sum_{i \in [m]} \lambda_{t,i} \left(V^{\pi_t, \widehat{P}_t}(\underline{g}_{t,i}) - \alpha_i\right) + \frac{1}{L+1} \sum_{i \in [m]} \lambda_{t,i} \left(V^{\pi_t, \widehat{P}_t}(\underline{g}_{t,i}) - \alpha_i\right)$$

$$= \frac{L}{L+1} \sum_{i \in [m]} \lambda_{t,i} \left(V^{\pi_t, \widehat{P}_t}(\underline{g}_{t,i}) - \alpha_i\right) + \frac{1}{\rho} \sum_{i \in [m]} \left[V^{\pi_t, \widehat{P}_t}(\underline{g}_{t,i}) - \alpha_i\right]^+, \tag{11}$$

where Equation (11) holds by definition $\lambda_t$.

Employing similar steps to Theorem 1, it holds, with probability at least $1 - \mathcal{O}(\delta)$:

$$\sum_{t=1}^{T} \left(V^{\pi^*}(r) - \sum_{i \in [m]} \lambda_{t,i} \left(V^{\pi^*}(g_i) - \alpha_i\right)\right) - \sum_{t=1}^{T} \left(V^{\pi_t}(r) - \sum_{i \in [m]} \lambda_{t,i} \left(V^{\pi_t}(\underline{g}_{t,i}) - \alpha_i\right)\right)$$

$$\leq \widetilde{\mathcal{O}}\left(\frac{L^3 m}{\rho} |X| \sqrt{|A|T} + \frac{L^5 m}{\rho}\right).$$

Adding and subtracting the estimated violation the following inequality holds:

$$\sum_{t=1}^{T} \mathcal{L}_{r,G}(\pi^*, \lambda_t) - \sum_{t=1}^{T} \left(V^{\pi_t}(r) - \sum_{i \in [m]} \lambda_{t,i} \left(V^{\pi_t}(\underline{g}_{t,i}) - \alpha\right) \pm \sum_{i \in [m]} \lambda_{t,i} \left(V^{\pi_t, \widehat{P}_t}(\underline{g}_{t,i}) - \alpha_i\right)\right)$$

$$\leq \widetilde{\mathcal{O}}\left(\frac{L^3 m}{\rho} |X| \sqrt{|A|T} + \frac{L^5 m}{\rho}\right),$$

which by Hölder inequality and Lemma 14 implies:

$$\sum_{t=1}^{T} \mathcal{L}_{r,G}(\pi^*, \lambda_t) - \sum_{t=1}^{T} \left(V^{\pi_t}(r) - \sum_{i \in [m]} \lambda_{t,i} \left(V^{\pi_t, \widehat{P}_t}(\underline{g}_{t,i}) - \alpha_i\right)\right)$$

$$\leq \widetilde{\mathcal{O}}\left(\frac{L^3 m}{\rho} |X| \sqrt{|A|T} + \frac{L^5 m}{\rho}\right). \tag{12}$$

Thus we substitute Equation (11) in Inequality (12) to obtain,

$$\sum_{t=1}^{T} \mathcal{L}_{r,G}(\pi^*, \lambda_t) - \sum_{t=1}^{T} \left(V^{\pi_t}(r) - \frac{L}{L+1} \sum_{i \in [m]} \lambda_{t,i} \left(V^{\pi_t, \widehat{P}_t}(\underline{g}_{t,i}) - \alpha_i\right)\right.$$

$$\left. - \frac{1}{\rho} \sum_{i \in [m]} \left[V^{\pi_t, \widehat{P}_t}(\underline{g}_{t,i}) - \alpha_i\right]^+\right)$$

$$\leq \widetilde{\mathcal{O}}\left(\frac{L^3 m}{\rho} |X| \sqrt{|A|T} + \frac{L^5 m}{\rho}\right).$$

To get back to the Lagrangian function of the offline problem, we notice that:

$$\frac{L}{L+1} \sum_{i \in [m]} \lambda_{t,i} \left(V^{\pi_t, \widehat{P}_t}(\underline{g}_{t,i}) - \alpha_i\right)$$

$$\geq \frac{L}{L+1} \sum_{i \in [m]} \lambda_{t,i} \left( V^{\pi_t, \widehat{P}_t} \left( g_i - 2\xi_t \right) - \alpha_i \right)$$

$$\geq \frac{L}{L+1} \sum_{i \in [m]} \lambda_{t,i} \left( V^{\pi_t}(g_i) - \alpha_i \right) - \frac{2Lm}{\rho} V^{\pi_t}(\xi_t) - \frac{2Lm}{\rho} \sum_{x \in X, a \in A} \left| \mathbb{P}_{\pi_t, \widehat{P}_t}(x,a) - \mathbb{P}_{\pi_t, P}(x,a) \right|,$$

and we employ Lemma 11 Lemma 14 to obtain:

$$\sum_{t=1}^{T} \mathcal{L}_{r,G}(\pi^*, \lambda_t) - \sum_{t=1}^{T} \mathcal{L}_{r,G}(\pi_t, {}^{\lambda_t L}/_{L+1}) + \frac{1}{\rho} \sum_{t=1}^{T} \sum_{i \in [m]} \left[ V^{\pi_t, \widehat{P}_t}(\underline{g}_{t,i}) - \alpha_i \right]^+$$
$$\leq \widetilde{\mathcal{O}} \left( \frac{L^3 m}{\rho} |X| \sqrt{|A|T} + \frac{L^5 m}{\rho} \right).$$

Thus, similarly to the steps above, we notice that:

$$\frac{1}{\rho} \sum_{i \in [m]} \left[ V^{\pi_t, \widehat{P}_t}(\underline{g}_{t,i}) - \alpha_i \right]^+$$

$$\geq \frac{1}{\rho} \sum_{i \in [m]} \left[ V^{\pi_t, \widehat{P}_t} \left( g_i - 2\xi_t \right) - \alpha_i \right]^+$$

$$\geq \frac{1}{\rho} \sum_{i \in [m]} \left[ V^{\pi_t}(g_i) - \alpha_i \right]^+ - \frac{2m}{\rho} V^{\pi_t}(\xi_t) - \frac{2m}{\rho} \sum_{x \in X, a \in A} \left| \mathbb{P}_{\pi_t, \widehat{P}_t}(x,a) - \mathbb{P}_{\pi_t, P}(x,a) \right|.$$

Employing Lemma 11 and Lemma 14 we obtain:

$$\sum_{t=1}^{T} \mathcal{L}_{r,G}(\pi^*, \lambda_t) - \sum_{t=1}^{T} \mathcal{L}_{r,G}(\pi_t, {}^{\lambda_t L}/_{L+1}) + \frac{1}{\rho} \sum_{t=1}^{T} \sum_{i \in [m]} \left[ V^{\pi_t}(g_i) - \alpha_i \right]^+$$
$$\leq \widetilde{\mathcal{O}} \left( \frac{L^3 m}{\rho} |X| \sqrt{|A|T} + \frac{L^5 m}{\rho} \right),$$

which can be rewritten as:

$$\frac{1}{\rho} \sum_{t=1}^{T} \sum_{i \in [m]} \left[ V^{\pi_t}(g_i) - \alpha_i \right]^+$$
$$\leq \widetilde{\mathcal{O}} \left( \frac{L^3 m}{\rho} |X| \sqrt{|A|T} + \frac{L^5 m}{\rho} \right) + \sum_{t=1}^{T} \mathcal{L}_{r,G}(\pi_t, {}^{\lambda_t L}/_{L+1}) - \sum_{t=1}^{T} \mathcal{L}_{r,G}(\pi^*, \lambda_t).$$

Thus, we employ Lemma 9 with Lemma 11 and Lemma 14 to obtain:

$$\frac{1}{\rho} \sum_{t=1}^{T} \sum_{i \in [m]} \left[ V^{\pi_t}(g_i) - \alpha_i \right]^+ \leq \widetilde{\mathcal{O}} \left( \frac{L^3 m}{\rho} |X| \sqrt{|A|T} + \frac{L^5 m}{\rho} \right),$$

from which we obtain the final bound:

$$V_T := \max_{i \in [m]} \sum_{t=1}^{T} \left[ V^{\pi_t}(g_i) - \alpha_i \right]^+$$

$$\leq \sum_{t=1}^{T} \sum_{i \in [m]} \left[ V^{\pi_t}(g_i) - \alpha_i \right]^+$$

$$= \rho \cdot \frac{1}{\rho} \sum_{t=1}^{T} \sum_{i \in [m]} \left[ V^{\pi_t}(g_i) - \alpha_i \right]^+$$

$$\leq \widetilde{\mathcal{O}} \left( L^3 m |X| \sqrt{|A|T} + L^5 m \right),$$

with probability at least $1 - \mathcal{O}(\delta)$, by Union Bound. This concludes the proof. □

# D  TECHNICAL LEMMAS

## D.1  POLICY OPTIMIZATION WITH DILATED BONUSES

In this section, we present the regret bound attained by PO-DB.

**Lemma 12** (Luo et al. (2021)). *For any sequence of losses $\ell_t$ such that $\ell_t \in [0, 1]^{|X \times A|}$ and any valid occupancy measure $\pi \in \Pi$, PO-DB attains:*

$$\sum_{t=1}^{T} V^{\pi_t}(\ell_t) - \sum_{t=1}^{T} V^{\pi}(\ell_t) \leq \widetilde{\mathcal{O}}\left(L^2 |X| \sqrt{|A|T} + L^4\right),$$

*with probability at least $1 - \mathcal{O}(\delta)$.*

## D.2  TRANSITION ESTIMATIONS

In the following section, we show how the estimated state action visit $\mathbb{P}_{\pi_t, \widehat{P}_t}(\cdot)$ concentrates to the true one $\mathbb{P}_{\pi_t, P}(\cdot)$.

To do so, we first provide some discussion on the transitions confidence set.

### D.2.1  CONFIDENCE SET

We introduce *confidence sets* for the transition function of a CMDP, by exploiting suitable concentration bounds for estimated transition probabilities. By letting $M_t(x, a, x')$ be the total number of episodes up to $t \in [T]$ in which the state-action pair $(x, a) \in X \times A$ is visited and the environment evolves to the new state $x' \in X$, we define the estimated transition probability at $t$ for the triplet $(x, a, x')$ as $\widehat{P}_t(x' \mid x, a) = \frac{M_t(x, a, x')}{\max\{1, N_t(x, a)\}}$. Then, the confidence set at $t \in [T]$ is $\mathcal{P}_t := \bigcap_{(x, a, x') \in X \times A \times X} \mathcal{P}_t^{x, a, x'}$, where:

$$\mathcal{P}_t^{x, a, x'} := \left\{\overline{P} : \left|\overline{P}(x'|x, a) - \widehat{P}_t(x'|x, a)\right| \leq \epsilon_t(x, a, x')\right\},$$

with $\epsilon_t(x, a, x')$ equal to:

$$2\sqrt{\frac{\widehat{P}_t(x'|x, a) \ln\left(\frac{T|X||A|}{\delta}\right)}{\max\{1, N_t(x, a) - 1\}}} + \frac{14 \ln\left(\frac{T|X||A|}{\delta}\right)}{3 \max\{1, N_t(x, a) - 1\}},$$

for some confidence parameter $\delta \in (0, 1)$.

The next lemma establishes $\mathcal{P}_t$ is a proper confidence set.

**Lemma 13** (Jin et al. (2020)). *Given a confidence parameter $\delta \in (0, 1)$, with probability at least $1 - 4\delta$, it holds that the transition function $P$ belongs to $\mathcal{P}_t$ for all $t \in [T]$.*

### D.2.2  CONCENTRATION RESULTS

Given the confidence set of the transition, it is possible to derive the following lemma.

**Lemma 14** (Lemma 4, Jin et al. (2020)). *With probability at least $1 - 6\delta$, for any collection of transition functions $\{P_t^x\}_{x \in X}$ such that $P_t^x \in \mathcal{P}_t$, we have, for all $x$,*

$$\sum_{t=1}^{T} \sum_{x \in X, a \in A} \left|\mathbb{P}_{\pi_t, \widehat{P}_t^x}(x, a) - \mathbb{P}_{\pi_t, P}(x, a)\right| \leq \mathcal{O}\left(L|X|\sqrt{|A|T \ln\left(\frac{T|X||A|}{\delta}\right)}\right).$$

As final remark, we underline that the empirical transition function $\widehat{P}_t$ belongs to $\mathcal{P}_t$ by construction. Thus, the aforementioned lemma immediately holds for $\widehat{P}_t$.

