# OpenReview forum: "Optimal Strong Regret and Violation in Constrained MDPs via Policy Optimization"
_ICLR.cc/2025/Conference — ICLR 2025 Poster_

### Official Review · Reviewer_QEvP · 2024-10-30

**Soundness:** 3
**Presentation:** 3
**Contribution:** 3
**Rating:** 6
**Confidence:** 3

**Summary:**

This paper studies learning constrained tabular MDPs with strong regret and violation guarantees. Prior works in this setting are either computationally inefficient or highly suboptimal. This work provides the first computationally efficient policy optimization algorithm with optimal $\sqrt{T}$ regret. The authors achieve this by leveraging the advance of adversarial MDPs for the primal update and an optimistic estimation for the dual update.

**Strengths:**

1. The problem studied in this paper is well-motivated and the strong regret/violation metric is reasonable.
2. The author successfully improves the regret bound in this setting from $T^{0.93}$ to the optimal $\sqrt{T}$ for computationally efficient algorithms. This is a huge improvement.
3. The writings are clear and discussion about previous works are sufficient.

**Weaknesses:**

1. This paper's algorithm and regret bound rely on a problem-dependent factor $\rho$, which could be small and lead to worse regret.
2. This paper does not have an empirical comparison. Although this is typically not necessary for a theoretical paper, simulation results like Muller et al. [2024] could be helpful.
3. A conclusion and discussion section is lacking.

**Questions:**

1. Could the authors provide more technical reasons why $\rho$ is required in this paper? Does this factor also appear in previous papers?
2. Is there any regret lower bound in this setting that is related to the number of constraints $m$?

---

> ### Author Response · Authors · 2024-11-15
>
> > This paper's algorithm and regret bound rely on a problem-dependent factor $\rho$, which could be small and lead to worse regret. Could the authors provide more technical reasons why $\rho$ is required in this paper? Does this factor also appear in previous papers?
>
> We thank the Reviewer for the opportunity to clarify this fundamental aspect. The dependence on $1/\rho$ is standard in primal-dual formulations (all the works that are mainly related to our, have this kind of dependence, see e.g., [Efroni et al 2020] and [Müller et al. 2024]). Intuitively, it happens since the optimal Lagrange variable of the offline problem is of the order $1/\rho$, and, the magnitude of the Lagrangian variables appears in the theoretical bound of primal-dual procedure.
>     To better understand this, notice that any regret minimizer scales at least linearly in its payoffs range. Thus, since the payoffs range of the primal depends on the maximum of Lagrangian variable, this dependence appears in theoretical bounds.
>
> Nonetheless, differently from [Müller et al 2024], we avoid the $1/\rho$ dependence in the violation bound, while keeping it in the regret only. We believe that this additional result is of particular interest for the community.
>
> Since this is a crucial aspect of our work, please let us know if further discussion is needed.
>
> > This paper does not have an empirical comparison. Although this is typically not necessary for a theoretical paper, simulation results like Muller et al. [2024] could be helpful.
>
> We agree that experiments are always beneficial; nevertheless, we underline that in the online CMDPs literature, many works do not have experimental results (e.g., Efroni et al. (2020)).
>
> > A conclusion and discussion section is lacking.
>
> We thank the Reviewer for the suggestion. We will surely include a conclusion in the final version of the paper.
>
> > Is there any regret lower bound in this setting that is related to the number of constraints $m$?
>
> We are not aware on any lower-bound related to the number of constraints in our setting. Nevertheless, to the best of our knowledge, there are not works which avoid the linear dependence on it, as in our work.

---

> > ### Comment · Reviewer_QEvP · 2024-11-25
> >
> > I thank the authors for the detailed response. I have no further questions and will keep my score.

---

### Official Review · Reviewer_fbBG · 2024-11-04

**Soundness:** 2
**Presentation:** 2
**Contribution:** 2
**Rating:** 6
**Confidence:** 4

**Summary:**

This paper studies online learning in constrained MDPs with strong regret and strong violation, where the negative terms are not allowed to compensate for positive ones. For this problem, this paper’s algorithm uses a primal-dual approach with UCB-like updates on the dual variables. The method achieves optimal $O(\sqrt{T})$ strong regret/violation, which improves the $O(T^{0.93})$ bound in the state-of-the-art works.

**Strengths:**

+ The concept of strong constraint violation is more relevant for safe-critical applications. It is also more challenging and technical than the conventional violation.

+ The paper proposed a primal-dual algorithm with an interesting dual design. The theoretical performance on regret and violation is also provided.

**Weaknesses:**

- The paper missed a few important related references (e.g., [1] and [2]), where the strong violation has been investigated and better results than this paper have been established. In [1], the OptPess-LP algorithm can satisfy the constraints instantaneously, which seems better than $O(\sqrt{T})$ strong violation.  In [2], the paper proposed a model-free method to achieve $O(\sqrt{T})$ strong violation. It would be better to discuss these papers in detail and highlight the differences.

- The algorithm requires Slater's condition and the knowledge of Slater's constant $\rho$, which is usually not practical in most critical applications. Besides, the regret is in the order of $O(1/\rho),$ it could be problematic when $\rho$ is close to zero.

- I understand it is a theory paper; however, including numerical experiments to validate the proposed algorithm would be beneficial. For example, the baselines could be Efroni et al. (2020), [1] and [2].

[1] Tao Liu, Ruida Zhou, Dileep Kalathil, PR Kumar, and Chao Tian. Learning policies with zero or bounded constraint violation for constrained MDPs. NeurIPS 2021.

[2] Arnob Ghosh, Xingyu Zhou, and Ness Shroff. Towards Achieving Sub-linear Regret and Hard Constraint Violation in
Model-free RL. AISTATS 2024.

**Questions:**

Please see the weakness.

---

> ### Author Response · Authors · 2024-11-15
>
> > The paper missed a few important related references (e.g., [1] and [2]), where the strong violation has been investigated and better results than this paper have been established. In [1], the OptPess-LP algorithm can satisfy the constraints instantaneously, which seems better than $O(\sqrt{T})$ strong violation. In [2], the paper proposed a model-free method to achieve $O(\sqrt{T})$ strong violation. It would be better to discuss these papers in detail and highlight the differences.
>
> We thank the Reviewer for the suggestion. Indeed, we will include both the papers mentioned by the Reviewer in the final version of the paper, including the following discussion.
>
> Specifically, while [1] studies CMDPs, their **OptPess-LP** algorithm **assumes the knowledge of a strictly feasible solution**. This assumption is non-practical in many real-world scenarios where the constraints are not known. Moreover, we remark that their approach is LP-based and not primal-dual, neither policy-optimization based.
>
> As concerns [2], as pointed out in [Müller et al, 2024], the algorithm proposed by the authors achieves $\widetilde{O}(\sqrt{T})$ strong violation when allowed to take $\Omega(d^{L-1}T^{1.5L}\log(|A|)^L)$ computational steps in every episode. Differently, in our work, as in [Müller et al, 2024], **we focus on polynomial-time algorithms that achieve a strong regret and violation guarantee**.
>
> Since these aspects are crucial, please let us know if further discussion is needed.
>
> > The algorithm requires Slater's condition and the knowledge of Slater's constant $\rho$, which is usually not practical in most critical applications. Besides, the regret is in the order of $O(1/\rho),$ it could be problematic when $\rho$ is close to zero.
>
> As concerns the Slater's condition assumption and the knowledge of $\rho$, these requirements are standard for primal-dual methods. Indeed, both [Efroni et al. 2020] -- for the primal-dual algorithm -- and [Müller et al. 2024] make the aforementioned assumptions. Intuitively, the Slater's condition is necessary, since, otherwise, the otpimal Lagrangian variable could be unbounded, preventing any kind of regret guarantees for primal-dual methods. As concerns the knowledge of $\rho$, this can be easily replaced by any lower-bound on $\rho$, or estimated in an online fashion adding a preliminary estimation phase (see Castiglioni et al. 2022, "A Unifying Framework for Online Optimization with Long-Term Constraints").
>
> A similar reasoning also holds for the dependence on $O(1/\rho)$. We remark that this dependence is standard for primal-dual methods (see [Efroni et al. 2020], [Müller et al 2024]). Intuitively, it happens since the optimal Lagrange variable of the offline problem is in general of the order $1/\rho$, and, the magnitude of the Lagrangian variables appears in the theoretical bound of primal-dual procedure. Nonetheless, please notice that, **differently from [Müller et al 2024], we avoid the $1/\rho$ dependence in the violation bound**. We believe that this additional result is of particular interest for the community.
>
> > I understand it is a theory paper; however, including numerical experiments to validate the proposed algorithm would be beneficial. For example, the baselines could be Efroni et al. (2020), [1] and [2].
>
> We agree that experiments are always beneficial; nevertheless, we underline that in the online CMDPs literature, many works do not have experimental results (e.g., Efroni et al. (2020)).

---

> > ### Comment · Reviewer_fbBG · 2024-11-26
> >
> > Thank the authors for the response. It has addressed some of the comments. I think the dual design is interesting and novel. However, given the related work [1] https://arxiv.org/pdf/2106.02684, I still have concerns about the contribution.
> >
> > In [1], if I understand correctly, the paper assumed a safe policy $\pi_0$ to ensure "zero-violation". Without the knowledge of  $\pi_0$, it might return a sublinear $O(\sqrt{T})$ constraint violation (please check Lemmas 5.1 and 5.2) as it could use unsafe policy at most $O(\sqrt{T})$ rounds to learn the rewards, costs, and kernel.
> >
> > Regarding computational complexity, [1] involves solving linear programming (LP) problems, while this paper requires dynamic programming (DP) to estimate the value function and a policy optimization solver (PO-DB). It would be more convincing to include a more detailed computational analysis.
> >
> > I am willing to increase the rating if the concern above is resolved.

---

> > > ### Author Response · Authors · 2024-11-27
> > >
> > > We thank the Reviewer for the comment. **Please notice the [1] does not attain results in terms of strong regret and violations**. Indeed, in [1], both regret and violations are defined so that they allow for cancellations (see Section 2). Specifically, notice that, in the constraints violations definition, the $[\cdot]_+$ operator is applied outside the summation, which is almost equivalent to the weak definition.
> > > Sublinear **weak** regret and violations have already been achieved in many existing works (e.g., the primal-dual algorithms in [Efroni et al., 2020]), while the objective of our work is to develop the first primal-dual algorithm to attain **optimal in $T$ strong regret and violations**.
> > >
> > > It is hard to make an **exact** comparison between the time complexity of linear programming and that of (primal-dual) policy optimization. Indeed, there are many algorithms that solve linear programs in polynomial time. However, they have polynomial running time with **high exponents and coefficients**, and in practice it is common to use solvers with exponential worst-case running time, but working better empirically. On the other hand, policy optimization methods (and primal-dual algorithms) usually require linear or at most quadratic running time. The higher efficiency of policy optimization is acknowledged by previous works (e.g., ["Optimistic Policy Optimization with Bandit Feedback" 2020] and ["Policy Optimization in Adversarial MDPs: Improved Exploration via Dilated Bonuses", 2021] for policy optimization in unconstrained settings, and [Efroni et al., 2020] and [Müller at al. 2024] for primal-dual policy optimization methods in constrained settings).
> > >
> > > Since these are crucial aspects of our work, please let us know if further discussion is required.

---

> > > > ### Comment · Reviewer_fbBG · 2024-11-27
> > > >
> > > > Thank you for your response. If I understand correctly, there exist two different definitions of violation in Section 2 in [1]: one is anytime violation (related to "strong violation") and the other is the cumulative violation you mentioned.
> > > >
> > > > OptPess-LP algorithm in [1] can indeed achieve "zero strong violation" (please see "Zero constraint violation case" paragraph in Section 2, Theorem 3.1, and Lemma 5.1). Moreover, though [1] did not clarify the results for strong regret, I believe it is true because the "LP-style" algorithm can guarantee anytime performance, implying a strong regret performance as well. This can be seen from the regret decomposition in (13) and the corresponding proofs in Lemmas 5.2~5.4.
> > > >
> > > > For the complexity, I think using LP to solve MDP or CMPD is a very classical and standard method, the complexity could be analyzed in terms of the sizes of state and action spaces. I am not following your statement on "high exponents and coefficients".  As in my previous comment, this paper requires both dynamic programming (DP) to estimate the value function and a policy optimization solver (PO-DB), where both policy evaluation and optimization contribute to the complexity, which might not have the advantage compared to LP-based method. Besides, I think your mentioned papers made such statements for some reasons or with evidence. It would be more convincing to point out them explicitly.

---

> > > > > ### Author Response · Authors · 2024-11-27
> > > > >
> > > > > > If I understand correctly, there exist two different definitions of violation in Section 2 in [1]: one is anytime violation (related to "strong violation") and the other is the cumulative violation you mentioned.
> > > > > OptPess-LP algorithm in [1] can indeed achieve "zero strong violation" (please see "Zero constraint violation case" paragraph in Section 2, Theorem 3.1, and Lemma 5.1). Moreover, though [1] did not clarify the results for strong regret, I believe it is true because the "LP-style" algorithm can guarantee anytime performance, implying a strong regret performance as well. This can be seen from the regret decomposition in (13) and the corresponding proofs in Lemmas 5.2~5.4.
> > > > >
> > > > > We apologize with the Reviewer since we probably misunderstood the previous question. We believed that the Reviewer was referring to the second algorithm in [1] (that is, the primal-dual one), which indeed does not attain strong regret and violations. As concerns the first algorithm, we believe it is possible to modify it to attain strong regret and violations. Nevertheless, notice that this modification would make OptPess-LP almost equivalent to the LP-based algorithms proposed in [Efroni et al., 2020], which attain strong regret and violations. Thus, we believe that **such algorithms' theoretical guarantees do not weaken our contribution, since they are not primal-dual**.
> > > > >
> > > > > > For the complexity, I think using LP to solve MDP or CMPD is a very classical and standard method, the complexity could be analyzed in terms of the sizes of state and action spaces. I am not following your statement on "high exponents and coefficients". As in my previous comment, this paper requires both dynamic programming (DP) to estimate the value function and a policy optimization solver (PO-DB), where both policy evaluation and optimization contribute to the complexity, which might not have the advantage compared to LP-based method. Besides, I think your mentioned papers made such statements for some reasons or with evidence. It would be more convincing to point out them explicitly.
> > > > >
> > > > > Regarding complexity, we agree that solving CMDPs has been originally tackled by employing LPs. Nevertheless, the following two considerations are in order. First, to the best of our knowledge, there are no works that study the exact time complexity of LPs to solve (optimistic) CMDPs. This is due to the fact that it is only possible to state that these optimistic LPs generally require $\mathcal{O}(|X|^2|A|)$ decision variables, while the exact complexity depends on the solver employed, which may lead to a time complexity scaling as $C (|X|^2 |A|)^d$ for large $C > 1$ and $d > 2$.
> > > > > Second, there is an extended literature which tries to solve CMDPs employing primal-dual methods, to reduce complexity. For instance, **[1] clearly states that "The OptPess-PrimalDual algorithm avoids linear programming and its attendant complexity and exploits the primal-dual approach for solving a CMDP problem. The proposed approach improves the computational tractability"** (Page 2, point 2), when comparing the first algorithm the Reviewer is referring to with their primal-dual procedure. Similarly, [Efroni et al., 2020] states "in the limit of large state space, solving such linear program is expected to be prohibitively expensive in terms of computational cost. Furthermore, most of the practically used RL algorithms are motivated by the Lagrangian formulation of CMDPs. Motivated by the need to reduce the computational cost, we follow the Lagrangian approach to CMDPs in which the dual problem to CMDP is being solved" (Section 5).
> > > > >
> > > > > Please let us know if you need further details.

---

> > > > > > ### Comment · Reviewer_fbBG · 2024-12-02
> > > > > >
> > > > > > Thanks for your response. I am still confused about the time complexity. Suppose we use interior-point methods; then the complexity of LP for (optimistic) CMDP should be $d=3$. This complexity seems no larger than your method, as your paper requires both dynamic programming (DP) to estimate the value function and a policy optimization solver (PO-DB).
> > > > > >
> > > > > > But I do think your policy-based method with function approximation has value and small computational complexity compared to the LP-based method.

---

> > > > > > > ### Author Response · Authors · 2024-12-02
> > > > > > >
> > > > > > > Following the example of the Reviewer, that is, employing the interior-point method with $d=3$ (and omitting for simplicity the dependence on $m$ and $L$), this would lead to a time complexity for LP-based methods of order $\mathcal{O}(|X|^6|A|^3)$; since the $d$ exponent is applied to the number of variables and constraints. Moreover, please notice that, while we can state the worst-case time complexity of LP-based methods, LP solvers often experience significant overhead due to the global nature of the optimization problem and face numerical instability due to ill-conditioned constraint matrices, especially when handling CMDPs with many states and actions.
> > > > > > >
> > > > > > > For our algorithm, the time complexity is of order $\mathcal{O}(|X||A|+C_{adv})$ where $C_{adv}$ is the time complexity of \texttt{PO-DB}, which is of order $\mathcal{O}(|X|^3|A|)$, that is, arguably better than the one of LP-based methods, and comparable to the time complexity of existing primal-dual methods (e.g., the one proposed in [1]). Moreover, notice that the time complexity of \texttt{PO-DB} may be eased by parallelization techniques, which is indeed not possible in linear programming.
> > > > > > >
> > > > > > > Furthermore, notice the following key feature of our algorithm. Precisely, our primal-dual scheme is independent from the specific policy-optimization procedure employed. We employed \texttt{PO-DB} because it is state-of-the-art in terms of efficiency. Nonetheless, it is possible to substitute it with any adversarial MDPs regret minimizer. Therefore, if future research develops a regret minimizer for adversarial MDPs with a time complexity of $\mathcal{O}(|X||A|)$, it can be directly incorporated into our framework without modification.
> > > > > > >
> > > > > > > We surely agree with the Reviewer that our technique is more akin to function approximation than LP-based methods. That is one of the reasons why primal-dual algorithms are preferred to LP-based ones and so heavily studied by the RL community.
> > > > > > >
> > > > > > > Finally, notice that, besides the time complexity, **our work answers a fundamental question in the RL theory community, that is, whether primal-dual methods may achieve optimal strong regret and violations. We believe this result is of interest for the community**.
> > > > > > >
> > > > > > > Please let us know if further clarification is necessary.

---

> > > > > > > > ### Comment · Reviewer_fbBG · 2024-12-02
> > > > > > > >
> > > > > > > > Thanks for your detailed response. I will increase my rating.

---

### Official Review · Reviewer_sekc · 2024-11-08

**Soundness:** 2
**Presentation:** 3
**Contribution:** 2
**Rating:** 6
**Confidence:** 2

**Summary:**

The paper studies an online learning problem in constrained MDPs. The authors propose a new constrained online learning algorithm that leverages an existing unconstrained policy optimization oracle. The authors prove that this method has the optimal regret and constrained violation bounds in a strong sense. This improves the state of the art bound of online learning in constrained MDPs.

**Strengths:**

- It is crucial to characterize stronger regret and constraint violation in online constrained MDPs since the transitional average performance metrics may obscure policy cancellation that is not permitted in safe policy learning.

- The authors provide an optimal regret and constraint violation bound by only considering the non-negative terms. This improves the previous suboptimal bound in the online constrained learning setting.

- The authors propose a new primal-dual online learning algorithm, which is different from the previous work that studies the strong regret and constraint violation. Rather than using regularization, the authors introduce several changes to the standard primal-dual methods: (1) binary dual update; (2) synthetic loss for policy optimization; (3) optimize policy through an existing adversarial policy optimization oracle.

**Weaknesses:**

- The authors focus on the basic tabular case of constrained MDPs. This method needs further generalization to extend beyond the tabular case.

- It would be helpful if the authors could clarify the motivation behind the techniques used in the proposed algorithm. Notably, the standard primal-dual policy optimization suffers the oscillation issues, potentially causing linear strong regret and constraint violation.

- The proposed algorithm employs an existing adversarial policy optimization oracle to update the policy. The policy optimization oracle is designed in the adversarial setting, while the constrained MDP problem assumes stochastic rewards, costs, and fixed transitions. It would be helpful if the authors could explain the rationale behind this choice.

- The adversarial policy optimization oracle minimizes the average type regret. It would be helpful if the extra technique to obtain a tighter regret bound can be highlighted.

- To illustrate the practical utility and verify the algorithm's performance, it would be helpful if the authors provided experimental results.

**Questions:**

- What is the role of the probability distributions in line 129 in algorithm?

- How large the margin $\rho$ is? What is the practical implication when it is infinitely small?

- Is it efficient to run the adversarial policy optimization oracle?

- Can the authors point out the new analysis that avoids the oscillation issue in typical primal-dual methods or compare their key analysis ideas?

---

> ### Author Response · Authors · 2024-11-15
>
> > The authors focus on the basic tabular case of constrained MDPs. This method needs further generalization to extend beyond the tabular case.
>
> We agree with the Reviewer that extending our results to non-tabular MDPs is an interesting future work. Nevertheless, we underline that, since the problem tackled by this work has been originally raised by [Efroni et al. (2020)], no better results than $\widetilde{\mathcal{O}}(T^{0.93})$ regret and violation have be shown even in the tabular setting. Thus, we believe that our result (which improves the bounds to $\widetilde{\mathcal{O}}(\sqrt{T})$) is still of fundamental importance for the community.
>
> >It would be helpful if the authors could clarify the motivation behind the techniques used in the proposed algorithm. Notably, the standard primal-dual policy optimization suffers the oscillation issues, potentially causing linear strong regret and constraint violation.
>
> We thank the Reviewer for the precious observation and for the opportunity to clarify this aspect of our work. The main advantage of our algorithm compared to existing primal-dual methods is that we somehow fix the Lagrangian variable depending on the policy chosen by the primal algorithm. To better understand this aspect, imagine a Markov decision process where the reward, the constraints and the transitions are fixed and known to the learner. Thus, we run a primal-dual procedure which does not employ any upper/lower confidence bound on the unknown variables (since all of them are known). In such a setting, standard primal-dual methods work by iteratively playing a Lagrangian game between the primal variable (the policy) and the dual ones: This leads to instability since no-regret adversarial procedures (employed for both the primal and the dual) would cycle around the equilibrium (namely, the optimal solution), as known from many results in equilibrium computation theory (see, e.g.,  "Prediction, learning, and games", Cesa-Bianchi and Lugosi 2006). On the contrary, our primal-dual scheme does not allow the Lagrangian variable to move at a rate of $1/\sqrt{T}$ (as for any primal-dual method which employ an adversarial regret minimizer for the dual), but it simply chooses between the maximum reasonable Lagrangian variable and the minimum one.
>
> We will surely include this discussion in the final version of the paper. Since it is a crucial aspect of our work, please let us know if further explanation is necessary.
>
> > The proposed algorithm employs an existing adversarial policy optimization oracle to update the policy. The policy optimization oracle is designed in the adversarial setting, while the constrained MDP problem assumes stochastic rewards, costs, and fixed transitions. It would be helpful if the authors could explain the rationale behind this choice.
>
> In primal-dual methods, it is standard to employ adversarial regret minimizers since the Lagrangian loss is adversarial for both the primal and the dual. To understand this, notice that, even if the rewards, the constraints and the transitions were deterministic, the Lagrangian function encompasses both the policy and the Lagrangian variables, which are selected by the algorithm and thus adversarial by construction.
>
> Please let us know if further discussion is necessary.
>
> > The adversarial policy optimization oracle minimizes the average type regret. It would be helpful if the extra technique to obtain a tighter regret bound can be highlighted.
>
>  We thank the Reviewer for the insightful comment and for the opportunity to clarify this aspect. It is important to underline that the average type of regret is computed w.r.t. the loss given to the primal algorithm. Notice that, this loss is built as the Lagrangian function employing upper bound on the rewards and lower bound on the constraints. Thus, we can see the loss as somehow deterministic, up to confidence bounds term (which shrinks sublinearly) and up to the Lagrangian variables, which are properly selected to make the primal avoid violations. Then, notice that average type of regret on deterministic functions coincides with the positive one, since it is not possible to perform better than the offline optimum.
>
> Since this is a crucial aspect, please let us know if further discussion is necessary.
>
> > What is the role of the probability distributions in line 129 in algorithm?
>
>  We are not sure to have properly understood the question. The reward and constraint distributions in line 129 are the ones that generate the feedback for our algorithm, that is, at each episode the learner observes a sample from those distributions for the path traversed in the CMDP.

---

> ### Author Response · Authors · 2024-11-15
>
> > How large the margin $\rho$ is? What is the practical implication when it is infinitely small?
>
> $\rho\in[0,L]$. When $\rho$ is arbitrary it may worsen the regret bound of our algorithm. Nonetheless, it is fundamental to take into account two crucial aspects. First, the dependence on $\rho$ for both regret and violation is standard in primal-dual formulations (all the works that are mainly related to our have this kind of dependence, see, e.g., [Efroni et al 2020] and [Müller et al. 2024]). Intuitively, it happens since the optimal Lagrange variable of the offline problem is in general of the order $1/\rho$, and, the magnitude of the Lagrangian variables appears in the theoretical bound of primal dual-procedure. Second, differently from [Müller et al 2024], we avoid the $1/\rho$ dependence in the violation bound. We believe that this additional result is of particular interest for the community.
>
> > Is it efficient to run the adversarial policy optimization oracle?
>
> We thank the Reviewer for the opportunity to clarify this aspect. The efficiency is one of the key advantages of our adversarial policy optimization procedure. Indeed, the update can be performed by employing dynamic programming techniques. Being a policy optimization approach allows to avoid the employment of occupancy-measure based methods which require projections to be performed at each episode (which, on the contrary, are highly inefficient). We refer to [Luo et al., 2021. "Policy Optimization in Adversarial MDPs: Improved Exploration via Dilated Bonuses"] for further details.
>
> > Can the authors point out the new analysis that avoids the oscillation issue in typical primal-dual methods or compare their key analysis ideas?
>
> Please refer to the second answer.

---

> ### Comment · Reviewer_sekc · 2024-11-26
>
> Thank you for the response. Since the response addresses my concerns, I am inclined to raise my score.

---

### Official Review · Reviewer_v2RE · 2024-11-12

**Soundness:** 3
**Presentation:** 2
**Contribution:** 2
**Rating:** 6
**Confidence:** 3

**Summary:**

This paper studies efficient online policy optimization in "*loop-free*" constrained MDPs (CMDPs) that slightly generalizes finite-horizon episodic CMDPs, where by "efficient" it refers to avoiding any optimization over the space of occupancy measures. In the *bandit-feedback* setting, it proposes $\texttt{CPD-PO}$, a primal-dual policy optimization algorithm built upon $\texttt{PO-DB}$ that achieves $\tilde{\mathcal{O}}(\sqrt{T})$ *strong* regret/violation bounds.

**Strengths:**

1. The paper studies a known open problem in literature that is of theoretical interest. The idea to consider *strong* versions of regret and constraint violation is reasonable and well justified.
2. The proof is checked to be correct, and the results do advance the theoretical understanding of policy optimization in CMDPs to a certain level.
3. I like Section 5.2 that compares the proposed algorithm against known algorithms.

**Weaknesses:**

1. This paper only deals with finite-horizon episodic MDPs with *bandit feedback*, which is a more restrictive setting than Efroni et al. (2020) and Müller et al. (2024). It seems a little unfair to directly compare against those algorithms that do not require bandit feedback.
2. The algorithmic contribution is limited since $\texttt{CPD-PO}$ largely builds upon $\texttt{PO-DB}$, only adding a simple binary dual update scheme.
3. Despite the theory-oriented approach of this paper, it is still helpful to include at least some simulation results to illustrate the applicability of the proposed algorithm.
4. The paper does not discuss about its limitations and future directions.
    * For example, the algorithm seems intractable since it requires the exact value of $\rho$, which is generally unavailable in practice. It would be much better if it can work with only an upper/lower bound of $\rho$, which does not seem to be the case here.
5. Suggestions on writing:
    * Avoid squeezing key formulations (i.e., the *loop-free* MDP setting) into the footnote, even given the page limit.
    * Clearly convey your message and ideas in the explanatory paragraphs following any mathematical results. For example, the paragraphs following Lemma 3 can be improved (What's the "aforementioned parameters"? Why does eq. (2) hold?).
    * The constants in Lemma 1 & 2 seem inconsistent from those in Lemma 6, up to a numerical factor.
6. Minor typesetting issues:
    * There are a few typos in the paper: $K$ should be $L$ in line 5 of Algorithm 2; missing $i$ in $i \in [m]$ in line 904; etc.
    * I would personally avoid using $\verb|\nicefrac|$ or anything similar to it because it makes fractions hard to read, esp. when you have something like $A+B+C / D+E+F$.

**Questions:**

Since loop-free MDPs are only a slight generalization of episodic MDPs, the dependency on $H$ also matters. Is $\tilde{\mathcal{O}}(L^5)$ the optimal dependency we can expect here (where $L$ is the horizon length)?

---

> ### Author Response · Authors · 2024-11-15
>
> > This paper only deals with finite-horizon episodic MDPs with bandit feedback, which is a more restrictive setting than Efroni et al. (2020) and Müller et al. (2024). It seems a little unfair to directly compare against those algorithms that do not require bandit feedback.
>
> We believe there is a possible **misunderstanding**.  Both [Efroni et al. 2020] and [Müller et al. 2024] **work under bandit feedback** as our paper, namely, observing the rewards and constraints along the path traversed during the episode. Notice that bandit feedback should not be seen as a strong requirement; indeed, it is standard in the online learning (and online RL) literature. To summarize, **our work closes the open problem raised by [Efroni et al. (2020)] and left open by [Müller et al. (2024)], for the setting studied by those specific works**. Since this is a crucial aspect of our work, please let us know if further discussion is necessary.
>
> > The algorithmic contribution is limited since $\texttt{CPD-PO}$ largely builds upon $\texttt{PO-DB}$, only adding a simple binary dual update scheme.
>
> We believe that this is indeed a point in favor of our algorithm. Indeed, any primal-dual algorithm employs an adversarial-kind of update for the primal (while dual methods employ UCB for the primal). This also holds for [Efroni et al. 2020] and [Müller et al. 2024], where the authors simply explicited the multiplicative-weight update. We decided to make our primal-dual scheme more general, namely, to rely on an existing algorithm for the primal regret minimizer, so that, in future, our primal-dual scheme could be instantiated with a different policy-optimization primal algorithm, in order to attain better regret and violations guarantees. Additionally, we believe this choice should improve the readability of our work.
>
> Nonetheless, we remark that the novelty of a primal-dual method generally relies on the specific Lagrangian formulation of the problem and on the primal-dual scheme employed. We believe that, since our primal-dual scheme is novel (e.g., employing UCB on the Lagrangian variable is a novel technique), the algorithmic novelty should be evaluated positively.
>
> > The algorithm seems intractable since it requires the exact value of $\rho$, which is generally unavailable in practice. It would be much better if it can work with only an upper/lower bound of $\rho$, which does not seem to be the case here.
>
> We thank the Reviewer for the opportunity to clarify this aspect. We first underline that the requirement on $\rho$ is standard in the literature of primal-dual methods (e.g., [Efroni et al. 2020] and [Müller et al. 2024]). Nonetheless, **our algorithm works for any lower-bound on $\rho$**. Indeed, it is possible to substitute $\rho$ with its lower-bound $\widehat{\rho}$, in Line 7 of Algorithm 2 (in the choice of $\lambda_t$) and all the results still hold, since, a greater Lagrangian variable does not preclude the possibility to achieve small violations. Nevertheless, please notice that, in such a case, the regret would scale as $1/\widehat{\rho}$. To conclude, we underline that there exist works which show how to introduce a preliminary estimation phase to estimate $\rho$ in an online fashion (see [Castiglioni et al. 2022, ``A Unifying Framework for Online Optimization with Long-Term Constraints"]).
>
> > Suggestions on writing and typos
>
>  We thank the Reviewer for the suggestions. We will surely include them in the final version of the paper.
>
> > Is $\tilde{\mathcal{O}}(L^5)$ the optimal dependency we can expect here (where $L$ is the horizon length)?
>
> We thank the Reviewer for the questions. We do not believe that the dependence on $L$ is tight. We leave as interesting open problem to develop an algorithm which attains tight regret and violation in any constant. Nevertheless, we believe that an improvement from $\widetilde{\mathcal{O}}(T^{0.93})$ to $\widetilde{\mathcal{O}}(\sqrt{T})$ should be evaluated positively.

---

> > ### Comment · Reviewer_v2RE · 2024-11-25
> > **Thanks for the responses.**
> >
> > The authors have made some good points in their rebuttal. I'm raising my rating to 6.

---

### Meta-Review · Area_Chair_FYQp · 2024-12-20

**Metareview:**

This paper proposes a novel algorithm in constrained MDPs that achieves strong regret and strong violation. The new algorithm is based on policy optimization and uses a primal-dual approach. The new result in this paper resolves an open question raised by prior work on this topic.

The theoretical contribution of this work could be of interest to the RL theory community. The reviewers also voted unanimously for acceptance.

**Additional Comments On Reviewer Discussion:**

The reviewers raised concerns regarding the feedback model, the algorithmic contribution of the paper and comparison with prior work. However, the authors provided detailed responses which successfully addressed those concerns and resulted in improved scores.

---

### Decision · Program_Chairs · 2025-01-22

Accept (Poster)